# Assessing the effectiveness of a national protected area network for carnivore conservation

J. Terraube[1 ✉], J. Van doninck [2], P. Helle[3] & M. Cabeza [1]

Protected areas (PAs) are essential to prevent further biodiversity loss yet their effectiveness varies largely with governance and external threats. Although methodological advances have permitted assessments of PA effectiveness in mitigating deforestation, we still lack similar studies for the impact of PAs on wildlife populations. Here we use an innovative combination of matching methods and hurdle-mixed models with a large-scale and long-term dataset for Finland's large carnivore species. We show that the national PA network does not support higher densities than non-protected habitat for 3 of the 4 species investigated. For some species, PA effects interact with region or time, i.e., wolverine densities decreased inside PAs over the study period and lynx densities increased inside eastern PAs. We support the application of matching methods in combination of additional analytical frameworks for deeper understanding of conservation impacts on wildlife populations. These methodological advances are crucial for preparing ambitious PA targets post-2020.

[1] Global Change and Conservation Lab, Organismal and Evolutionary Biology Research Program. Faculty of Biological and Environmental Sciences, University of Helsinki, PO Box 65 (Viikinkaari 1), FI-00014 Helsinki, Finland. [2] Amazon Research Team, Department of Biology, University of Turku, 20500 Turku, Finland. [3] Natural Resources Research Institute, Paavo Havaksen tie 3, FI-90570 Oulu, Finland. ✉email: jterraube@gmail.com

Protected areas (PA) are perceived as key conservation instruments[1–3] yet we know very little about their effectiveness at safeguarding biodiversity against increasing human pressures[4]. Given the utmost importance of these instruments for the protection of species[5], the maintenance and restoration of ecosystem functioning[6] and their contribution to human well-being and poverty alleviation[7,8], it appears as even more urgent to improve our understanding of the effectiveness of PAs worldwide. The last two decades have put more attention on PA efficiency (e.g., coverage of species ranges within protected areas[9]) than on ecological effectiveness[10]. While such appraisals have been important in driving ambitious conservation targets globally (Aichi Target 11 of the Convention on Biological Diversity http://www.cbd.int/sp/targets), calls for assessing the effectiveness of established and expanding PAs are increasing.

Recently, interest has shifted towards evaluations of Protected Area Management Effectiveness[11,12] and assessments of impact of PAs in reducing threats[3,13,14]. For this latter, efforts have been directed to the development of statistical approaches referred to as matching tools. These approaches account for confounding factors, trying to separate the effects of PA location (and thus pressures faced) and PA management (law enforcement or active management to improve habitat and populations). In this regard, studies applying matching tools to large scale and fine resolution deforestation data have become increasingly popular[15–17]. They have generally shown that PAs do have an effect in reducing deforestation rates although the effect is not as large as thought prior to applying such counterfactual methods[18].

Albeit not with matching approaches, some studies have addressed the effectiveness of PAs for protecting wildlife species. These studies found more positive wildlife population trends in sites located inside PAs compared to sites located outside[19], stable wildlife population trends inside tropical PAs[20] or an important contribution of PA networks to the protection of megafauna habitat in some countries[21]. However, the abovementioned deforestation studies warn that omission of counterfactuals could have led to an overestimation of PA impacts on wildlife. The application of matching approaches to wildlife populations has been lagging behind, partly because of lacking comprehensive datasets allowing to compare protected and non-protected sites of similar environmental characteristics. However, moving from analyses of deforestation to more complex conservation targets for wildlife populations comes with additional conceptual and methodological challenges.

Large carnivores are globally vulnerable to increasing anthropogenic pressures, with the populations of several iconic species, e.g., cheetah (*Acinonyx jubatus*) and African lion (*Panthera leo*), severely declining in large parts of their distribution range[22,23]. The degradation and loss of habitat in addition to human retaliatory persecution following livestock predation have historically driven a decline in most large carnivores globally. Therefore, understanding the outcomes of conservation actions for this group of species is pressing given the important ecological role of carnivores in regulating ecosystems and the profound significance of these species to people worldwide[24]. The evaluation of the impact of conservation actions is relevant in the European context where large carnivore populations are now recovering at a continental scale[25].

Although carnivores have large home-ranges and may often roam outside PAs, we expect land protection to have a positive effect on carnivore occurrence and population densities. In general, we expect PAs to reduce direct anthropogenic killing and increase the availability of resources in better quality habitat targeted by protection. Hunting of carnivores is allowed in Fennoscandia based on hunting quotas determined through population estimates. Yet the largest concern arises from illegal killing[26,27], observed even inside PAs[28] despite high levels of

governance in Fennoscandia´s countries[29]. In Finland, over the last three decades, the development of adaptive management plans and increased prey abundance have contributed to the recovery of large carnivore populations, although wolf population growth rates have been limited since 2006 as a result of widespread poaching undermining conservation efforts[30,31].

Here, we develop an application of PA effectiveness evaluation approaches taking advantage of a unique dataset arising from the Finnish Wildlife Triangle Scheme (FWTS), which has collected wildlife abundance data throughout Finland since 1989 in ~2000 12 km transects. Using this data we assess the effectiveness of Finland´s PA network in supporting abundances of four large carnivore species: Eurasian lynx (*Lynx lynx*), Gray wolf (*Canis lupus*), Wolverine (*Gulo gulo*), and Brown bear (*Ursus arctos*). We apply an innovative combination of matching methods and hurdle-mixed effects models to strengthen our inference on the effectiveness of PAs at protecting carnivore populations at the national scale.

We would expect higher population densities in protected transects compared to similar non-protected transects if PAs are effective in maintaining large carnivores. However, due to variations in ecological niche and conflict levels among the four study species, the strong latitudinal and longitudinal gradients in PA coverage or size, as well as regional differences in attitudes towards large predators, we predict that PA impacts will vary (1) between species at the national network scale with the wolverine (a specialist species[32] sensitive to interactive effects of climate and landscape change[33]) being the most likely species to be positively affected by PAs; (2) between regions with lower PA effectiveness expected in Southern Finland (where PAs are small) and in Lapland (as a result of conflicts with reindeer herders); (3) over the study period with a potential decline in PA effectiveness concomitant to increased levels of human-carnivore conflicts at the national level.

PAs do not have an effect on population densities for three (lynx, wolf and wolverine) of the four species at the national network level. Interestingly, the two different analytical approaches lead to contrasting effects of PAs on brown bear densities that we explore further. Moreover, spatial and temporal variations in PA effectiveness are revealed only when combining the two analytical approaches. Our results provide a first application of statistical matching methods to wildlife data. They highlight potential points for improvement to strengthen carnivore conservation at the national scale and will foment methodological progresses in PA impact assessments.

## Results and discussion

**Matching analyses reveal no PA effects on carnivores.** Finland has a rather uneven distribution of PAs (Fig. 1; Supplementary Table 1) but an extensive forest cover (76% of the country) that supports the persistence of carnivore species throughout. From the 1805 sampling units available, we were able to pair 1220 units (610 inside PAs and 610 outside PAs) using a matching procedure based on habitat and accessibility characteristics considered likely to influence large carnivore population trends. Most of these matched pairs of PA and non-PA sampling units had carnivore population data covering a period of ~30 years and represented well the four main biogeographical regions (n = 563 paired sampling units; total of 3111 observations for all units and all years, see Supplementary Table 2).

When all years and all regions were pooled, we did not find significant differences between densities of lynx (−0.077; 95% CI: −0.250, 0.0913; n = 207 units), wolverine (median absolute PA effect = 0.097; 95% CI: −0.250, 0.091; n = 75 units) and wolf (median absolute PA effect = 0.018; 95% CI: −0.031, 0.077; n = 67 units)

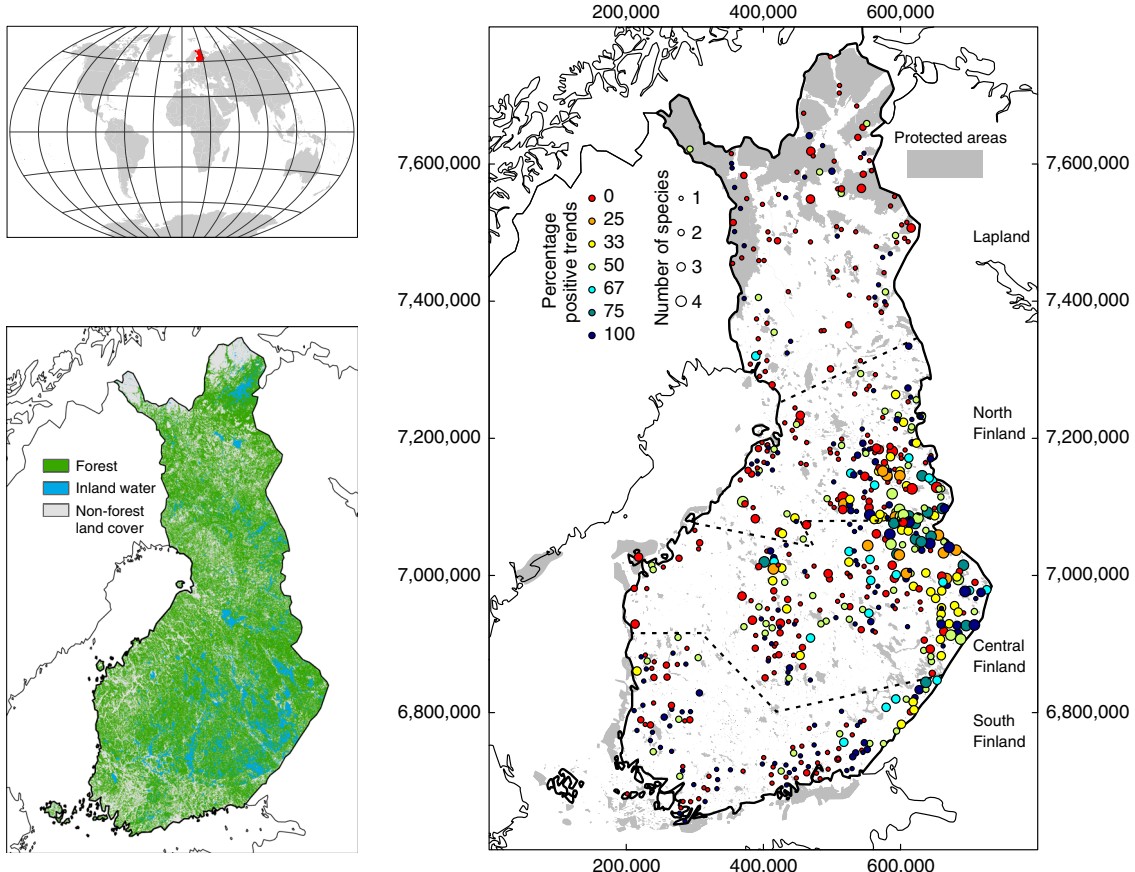

**Fig. 1 Carnivore population trends by sampling unit.** Maps showing (i) the location of Finland in the world (top left); (ii) the extent of forest cover at the national scale (lower left); and (iii) all monitored units integrated in the Finnish Wildlife Triangle Scheme (right), showing the percent of positive population trends between 1989 and 2017 depending on the number of large carnivore species detected at each site and their population trends at the respective sites (e.g., 1 species with positive trends and 2 species with negative trends: 33%). Colored circles represent monitored sampling units. Circle size represents the number of carnivore species detected while circle color represents the percent of positive population trends per unit. Game management areas are separated by dashed lines (Southern Finland; Central Finland; Northern Finland and Lapland) and gray areas represent the Finnish protected area network.

between sampling units located inside and outside protected areas. However, brown bear densities were lower inside protected areas than outside (median absolute PA effect = −0.311; 95% CI: −0.523, −0.081; $n$ = 209 units) (Supplementary Table 3).

There was no difference in densities between PAs and non-PAs for individual years in any of the carnivore species (Bear: median absolute PA effect range (−1.443; +1.453), all Bonferroni-adjusted $p$-values = 1; Lynx: median absolute PA effect range (−1.107; +0.847), all Bonferroni-adjusted $p$-values = 1; Wolf: median absolute PA effect range (−7.456; +6.309), all Bonferroni-adjusted $p$-values = 1 for years with number of observations>10; Wolverine: median absolute PA effect (−1.689; +1.281); all Bonferroni-adjusted $p$-values = 1 for years with number of observations>10; see Fig. 2 and Supplementary Table 4).

Similarly, regional matching analysis (separate analyses for southern Finland, central Finland, northern Finland and Lapland) did not reveal contrasted spatial patterns in PA performance for any species (all 95% confidence intervals overlap 0; see Supplementary Table 5).

**Combining approaches to corroborate effectiveness estimates.** For independent validation of matching outcomes and further understanding of fine-scale spatial and temporal patterns in effectiveness, we used density data to fit hurdle-mixed effects models for each species. This allowed us to assess how land protection influenced the response variable together with year

and 6 other environmental covariates that were also part of the initial matching process. This second approach confirmed overall the results obtained through the matching methods, with no global effect of protected areas detected for lynx (GLMMzi: estimate = 0.0041; $z$ = 0.6236; $p$ = 0.533), wolverine (GLMMzi-estimate = −0.0018; $z$ = −0.659; $p$ = 0.509) and wolf densities (GLMMzi-estimate = 0.0008; $z$ = 0.330; $p$ = 0.742) (Fig. 3; see Supplementary Table 6 for model selection results and Supplementary Table 7 for model estimates). However, the results for bear were opposite to those from matching, indicating higher brown bear densities inside protected areas than outside (GLMMzi-estimate = 0.0065; $z$ = 1.8754; $p$ = 0.0607; Fig. 3, see Supplementary Table 6 for model selection results and Supplementary Table 7 for model estimates).

These contrasting findings for bear densities are linked to two main factors: (i) the structural difference of the datasets used in both approaches, which led to a smaller number of (paired) samples in the matching approach (Supplementary Table 2; Supplementary Table 8); (ii) the different hierarchical scales associated to the matching methods and hurdle-mixed models. A negative absolute PA effect (outcome of matching) means that overall, at the unit level, bear densities are higher at a non-protected site than at a protected one of similar characteristics. Instead, a positive marginalized coefficient from the hurdle-mixed models means that, across the entire dataset, bear densities are higher at protected sites than at non-protected ones. High spatial

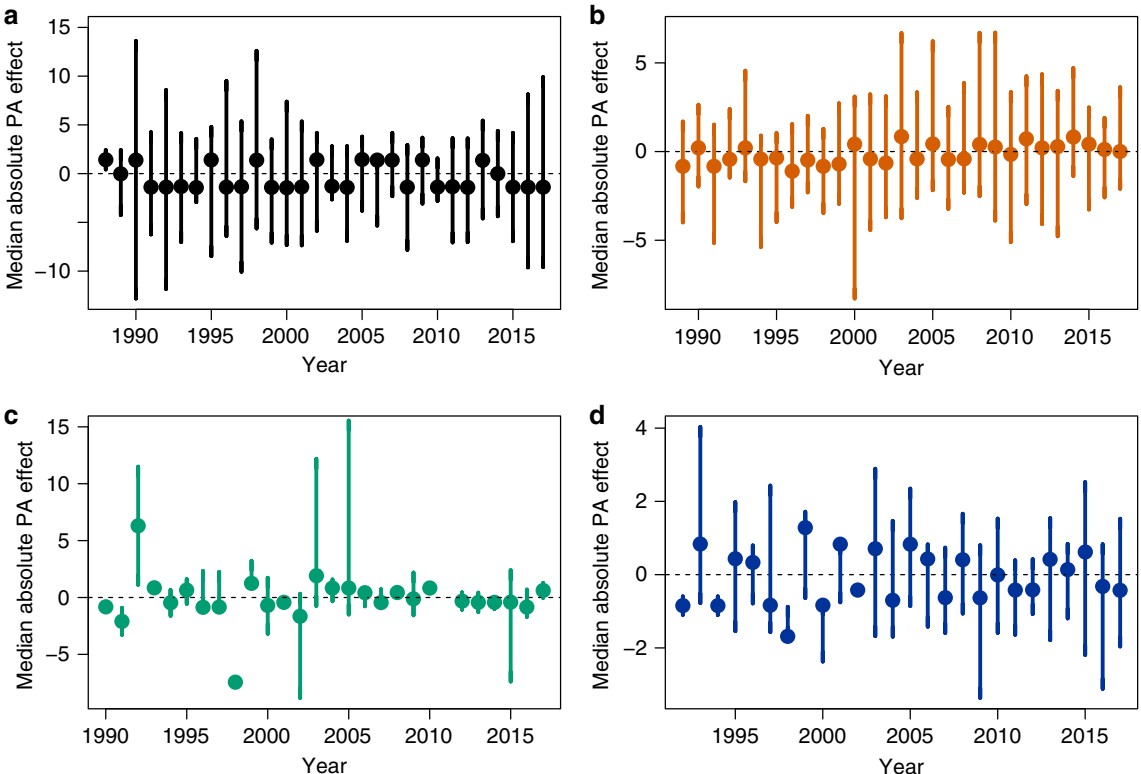

**Fig. 2 PA effectiveness estimates derived from matching analyses.** Mean annual absolute effect of the Finnish PA network on (**a**) bear ($n = 209$ paired sampling units), (**b**) lynx ($n = 207$), (**c**) wolf ($n = 67$) and (**d**) wolverine ($n = 75$) densities during 1989–2017, using only matched protected and unprotected wildlife units. Colored dots and arrows represent median estimates and 95% confidence interval, respectively.

heterogeneity in bear densities nationwide could potentially explain the pattern observed here as few protected sites with high bear densities could drive the global positive effect of PAs on bear densities in the hurdle mixed model outputs.

Overall, we found only small variation between species, only weakly supporting prediction 1. The different outcome for brown bear as opposed to the rest of the species could potentially be explained by dietary preferences. High ungulate abundance outside PAs[34] could explain the observed absence of PA effect on lynx, wolf, and wolverine densities, but the different ecological requirements of the omnivorous brown bear[35] make this species less dependent on high ungulate availability found in anthropogenic landscapes and could contribute to the differences found in effectiveness.

Additional differences between species become more apparent with the hurdle-mixed model, which better identifies variables interacting with protection. This is exemplified by the regional differences found for lynx or the effect of time for wolverine. We found some support for prediction 2 (lower PA effectiveness only consistent for Lapland, yet no pattern for Southern Finland and instead differences were detected between East and West) and only weak support for prediction 3 (decreasing effectiveness with time just for one of the species) that we discuss next. In relation to regional effects we detected an interactive effect of protection status and longitude on lynx densities, namely lynx densities were higher outside PAs than inside in western Finland but the opposite was observed at higher longitudes, i.e., in Eastern Finland (GLMMzi-estimate = −0.018; $z = −5.567$; $p = 0.010$; Fig. 4a). This could be related to higher prey availability inside PAs in this area (mountain hare (*Lepus timidus*) is the main prey of the boreal lynx in this part of the country[36]) associated to overall high habitat suitability and geographic proximity to the Russian source populations[37].

No effects of PAs were detected for wolverine using the matching methods (none of the annual median estimates of

absolute PA effect was significantly different from 0; Supplementary Table 4). However, the hurdle mixed model highlighted an interactive effect of protection status and years showing stable wolverine densities outside protected areas, but a declining trend inside PAs over the ~30-year study period (GLMMzi-estimate = −0.005; $z = −2.029$; $p = 0.042$; Fig. 4b).

Notably, in hurdle mixed models built without the Lapland sampling sites, the interaction between PAs and years disappears suggesting that the Lappish sites are the ones driving this interaction. An estimated total of around 1200 wolverines in the three Nordic countries, Norway, Sweden and Finland makes the wolverine the rarest of the four large carnivore species in the European Union, while it hosts the status of endangered in the Finnish Red List. Therefore, the decline in wolverine densities observed inside PAs is alarming given the responsibility of Nordic countries regarding wolverine conservation in a European context[38]. This could be related to a temporal increase in anthropogenic mortality inside PAs located in northern Finland and adds to previous evidence of higher poaching of large carnivores inside than outside PAs located in Swedish Lapland[28].

In Finland, permits are regularly granted to hunt large carnivores inside protected areas on the grounds of prevention of damages to livestock[39], as semi-domestic reindeers are a regular prey of wolverine and lynx in northern Fennoscandia[40]. Depredation incidents within the reindeer husbandry area in Lapland drive important human-carnivore conflicts, challenging large carnivore conservation in these areas[41]. In addition, spatial patterns of illegal killing may be mediated by habitat quality: semi-domestic reindeers may be more frequently found during winter inside northern protected areas as a result of high availability of grazing resources (lichens) in protected old-growth forests. This is possibly due to large-scale loss of old-growth forests outside PAs linked to intensive forestry practices[42]. Consequently, the overall pattern described

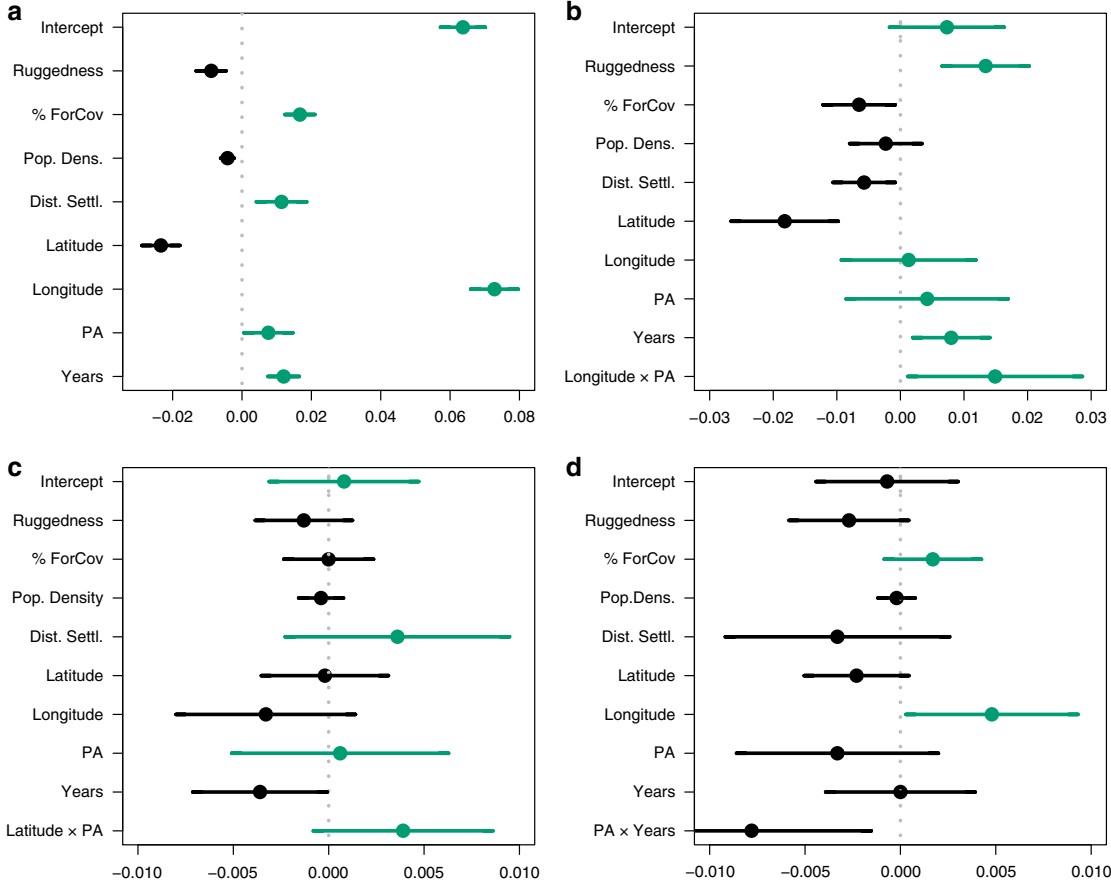

**Fig. 3 PA effectiveness estimates derived from hurdle-mixed models.** Marginal regression coefficients (colored dots) and 95% confidence intervals (arrows) from hurdle-mixed models for the impacts of PA (as well as additional matching covariates and interactions) on (**a**) bear, (**b**) lynx, (**c**) wolf, and (**d**) wolverine densities ($n = 1126$ unpaired sampling units). (Estimates of covariates having a positive effect on carnivore densities are represented in green and estimates of covariates having a negative effect on densities are represented in black).

here might trigger increasing reindeer attack rates inside PAs, worsening the human-carnivore conflict in Lapland[43,44].

In this sense, it appears urgent to develop and improve co-management of northern protected areas in order to alleviate conservation conflicts[45]. Higher PA effectiveness in northern Finland would increase transboundary connectivity for carnivores, which is crucial for their large-scale population dynamics across all Fennoscandia.

Given spatial variations in PA size at the national level (Supplementary Table 1), it is worth highlighting that PA size had no effect on the densities of the four species of large carnivores studied here and cannot be considered a confounding factor biasing our PA effectiveness estimates (GLMMzi: Bear-PA size: estimate = 0.0011; $z = 0.269$; $p = 0.788$; GLMMzi: Lynx-PA size: estimate = 0.0019; $z = 0.698$; $p = 0.484$; GLMMzi: Wolf-PA size: estimate = $-0.0032$; $z = -0.949$; $p = 0.342$; GLMMzi: Wolverine-PA size: estimate = $-0.0003$; $z = -0.104$; $p = 0.917$).

**Potential limitations.** We note that commonly used approaches to assess PA effectiveness have their caveats[46]. For instance, a lack of difference between densities in PA and non-PA sites does not necessarily indicate whether PAs are being ineffective, unless accounting for pressures separately. Paired sites can have similar densities for numerous reasons. For example, lower anthropogenic mortality inside PAs could be compensated by higher prey availability outside PAs[35]. Alternatively, benefits associated to higher habitat quality inside PAs could be annihilated by higher anthropogenic mortality inside PAs as previous research

has shown in Swedish Lapland[28]. Yet the latter would require prioritizing management actions in these poorly performing PAs.

Another potential caveat of our approach is linked to the fact that bear density data have been collected in a different season (end of summer) than data from the other three carnivores (mid-winter). As a result, we cannot rule out that the absence of PA effects on lynx, wolf and wolverine densities is related to seasonal variations in habitat selection with non-protected open habitats and sites with higher human presence being more used in winter by these species[47,48]. These seasonal variations can be related to the summertime rearing of pups which are less mobile and require protection[49] and changes in prey habitat selection and body condition between summer and winter[50]. Further research should quantify seasonal variations (summer vs. winter) in the use of PAs by these species through fine-scale GPS tagging.

Finally, we acknowledge the methodological challenges associated to impact evaluations of PA networks on highly mobile species, such as large carnivores, particularly using indirect indices of density. However, snow track counts have been used to monitor the density of several large carnivores in the northern hemisphere, including the wolverine[51], the gray wolf[52], and the lynx[53]. Therefore, we are confident about the robustness of the density indices derived from the snow track counts and their use in PA effectiveness assessments.

In conclusion, our results generally revealed a lack of differences in large carnivore densities between protected and non-protected sites in Finland. However, these outcomes should not be interpreted as if PAs were irrelevant for carnivore conservation but instead they

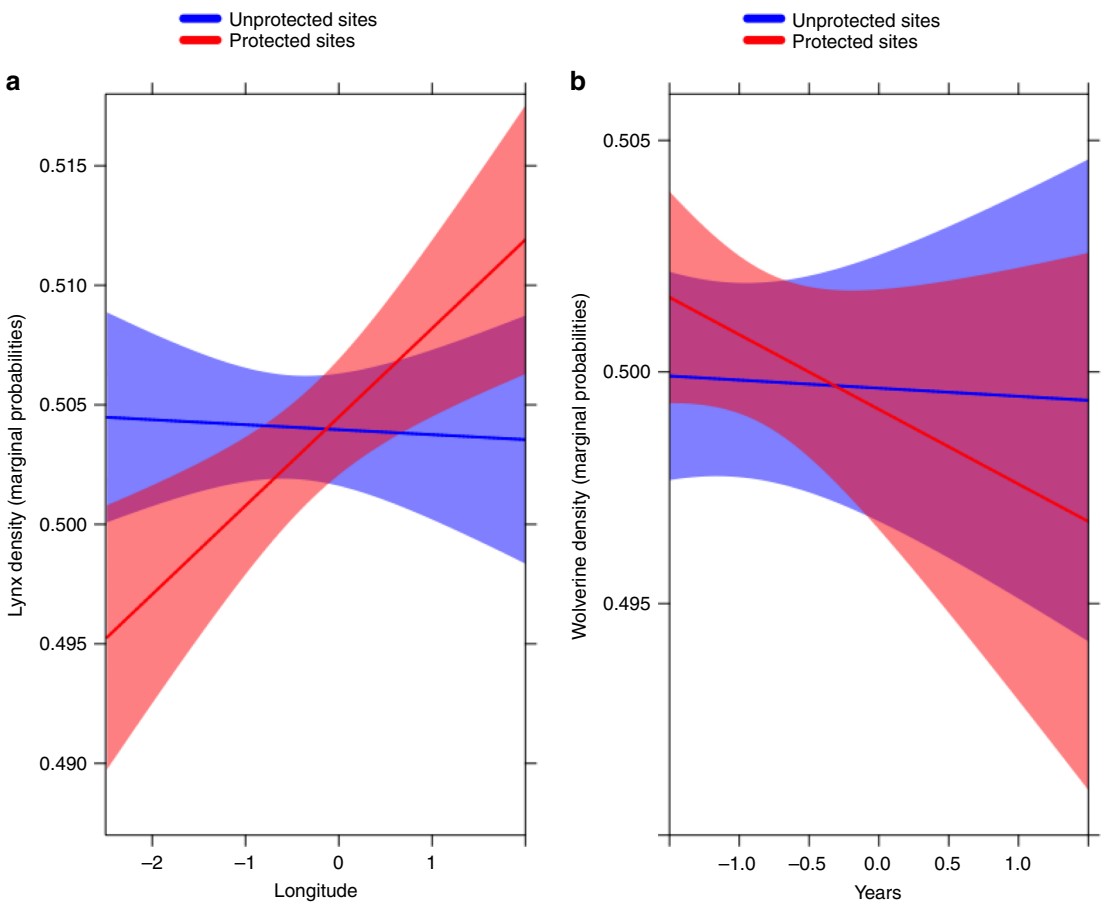

**Fig. 4 Spatio-temporal variation in effectiveness of the Finnish PA network.** Predicted values (solid lines) and 95% confidence intervals (shaded areas) of (**a**) lynx and (**b**) wolverine densities in relation to longitude and years, respectively, for sites included inside the Finnish PA network and sites located outside protected areas.

reveal complex patterns of interaction between law enforcement gaps, hunting pressure, and prey availability. To our knowledge, this is the first matching analysis applied to wildlife data. This was aimed at questioning whether increasingly used approaches to assess effectiveness in mitigating deforestation should also be used to evaluate wildlife trends. Although caveats in matching methods have also been denoted in the latter applications[46], their use is still preferred. When it comes to wildlife trends, we noted that data of the required extent will rarely be available. When available, our study has raised important considerations as we recommended caution in their application especially to assess highly mobile wildlife not confined within PAs. Yet, applied with long-term and large-scale data and in combination with other models, as done here, we believe matching approaches are a valid approach for wildlife population trends, serving as the basis for further research.

We believe that our findings will shed more light and foment further advances on the potential applications of matching methodologies for PA effectiveness assessments[54,55]. These methodological advances are essential as we should rapidly upgrade international targets concerning PA management and their ecological outcomes[56,57] if we aim to achieve the 2030 Agenda for Sustainable Development Goal 15 and halt global biodiversity loss.

## Methods

**Wildlife population time series data.** Track count data collected within the Finnish Wildlife Triangle Scheme were used as indices of lynx, wolf, wolverine and bear densities. The scheme is a long term, large-scale survey which provides yearly

estimates of the distribution and relative abundance of game species. The methods are described in detail by Linden et al.[58]. A wildlife unit is a permanent line transect route of 12 km (4 × 3 km). Unit locations represent different forest habitats in proportion to local occurrence. The monitoring operation is carried out twice a year. The minimum number of monitoring experts is three for the late-summer count and one for the winter count.

Each winter, 800–900 units distributed throughout Finland are surveyed by local hunters who count the number of fresh snow tracks crossing the transect. The count is either done 1–2 days after a snow fall or 24 h after a pre-check where all old tracks are marked. For each species, the data are compiled by the Finnish Game and Fisheries Research Institute as the number of crossings per 24 h per 10 km. Thus, the unit used in statistical modeling was the average number of crossings per 24 h per 10 km for lynx, wolf and wolverine, per triangle and for each year. As brown bears hibernate during winter counts, density indices for this species were collected during late summer counts and also focused on counting presence indices along transects.

From the initial 2171 sampling units, a total of 1805 units were included in subsequent analyses (we discarded transects in mixed agricultural-forest landscapes), 1195 were located in non-protected sites while 610 were located inside protected areas.

**Explanatory variable selection and extraction.** The set of matching covariates extracted for each wildlife unit represented ecological characteristics considered most likely to be important determinants of large carnivore occurrence in European ecosystems based on empirical observation[59]. These covariates can be regarded as two groups of possible influences: biophysical context (e.g., latitude) and human impacts (e.g., distance to closest settlements). Several variables suggested by the literature to be important in determining PA effectiveness, e.g., PA management, were available for a restricted amount of sites and therefore were not included in subsequent analyses. Description of the matching covariates and the rationale for their inclusion are found in the Supplementary Table 9.

Spatial data were analyzed using the R packages sp, raster, rgdal, and rgeos[60]. PA boundaries were calculated using spatial information from the World Database of

Protected Areas[61]. A sampling unit was considered as protected if the 1 km circular buffer centered on the unit's centroid intersected a protected area polygon. Other explanatory variables corresponding to each wildlife unit were extracted for the unit's centroid –for distance variables or gridded variables with a spatial resolution larger than the unit's dimensions—or over a circular buffers centered on the unit's centroid and passing through the three vertices—for gridded variables with a spatial resolution smaller than the unit's dimensions (Supplementary Table 9).

**Estimating protected area effectiveness with matching methods**. We used the Matchit package[62] in R environment, which fits a logistic generalized linear model where the treatment assignment (land protection) is the response variable and the matching variables are the predictors.

We chose one-to-one, nearest-neighbor covariate matching with replacement using a generalized version of the Mahalanobis distance metric. We used a caliper of 0.2 standard deviations of the propensity scores as our minimum matching criterion. To assess the quality of the matches, we checked the resulting covariate balance testing the normalized differences in covariate means and their distribution. The normalized difference in means is the difference in the average covariate value divided by its standard deviation[63]. We tested for differences in the distribution using eQQ plots that graph the covariate values in the same quantile of the treated (protected sites) against those in the control (non-protected sites), allowing us to observe if characteristics are distributed similarly across both treatment and control groups (see Supplementary Figs 1a and 1b).

We were able to match 100% of the original sample to controls that suited the criteria. Therefore, our unit-to-unit matching yielded a final dataset of 1220 sampling units, i.e., 610 protected wildlife units that were matched to 610 similar unprotected units across Finland. These matched pairs of PA and non-PA sampling units covering, a period of ~30 years, were included in subsequent analyses. These sampling units were not homogeneously distributed among the four Game Management clusters with Central Finland being the most represented cluster and Lapland the least (see Supplementary Table 2).

Previous studies have quantified the effect of PAs in reducing threats in different ways (Absolute PA effect, Relative PA effect, Pooled relative effect[17]). However, due to limitations imposed by the structure of wildlife time series and the natural low densities of large carnivores (zero inflation of all time-series), an effectiveness metric based on the 'absolute PA effect' was the most relevant approach for this study.

The absolute PA effect is the difference between densities in a unit located inside a PA and its matched control unit located in a non-protected area. Therefore a positive value means that sampling units located inside PA show higher large carnivore densities than its control unprotected units. We calculated this metric at the national PA network level, for each pair of 'protected-non protected' units. We computed the median absolute effect of the PA network and its associated 95% confidence interval for each species of large carnivore, globally and at the regional level. In both cases, we performed iterative random sampling (1000 iterations) to control for differences in the number of repeated observations (number of years) among pairs of matched units. This was done by selecting only one value of absolute PA effect for each pair of matched units before estimating the median PA effect across units. To calculate the absolute PA effect per region, we pooled the 15 Game Management Areas covering the whole country in four main clusters: South = 'Satakunta' + 'Etelä-Häme' + 'Kaakkois-Suomi' + 'Uusimaa' + 'Varsinais-Suomi' + 'Pohjois-Häme'; Central = 'Etelä-Savo' + 'Rannikko-Pohjanmaa' + 'Keski-Suomi' + 'Pohjois-Savo' + 'Pohjois-Karjala' + 'Pohjanmaa'; North = 'Oulu' + 'Kainuu'; Lapland = 'Lapland'. We compared the annual absolute PA effect per year with zero using one sample t-tests.

**Estimating PA effectiveness with two-part mixed effects model for semi-continuous data**. We extracted the matched dataset obtained through the matching process described previously (1220 unpaired wildlife units, see Supplementary Table 8) to test for the effect of land protection status on large carnivore densities. We implemented two-part zero-inflated mixed effects models in the novel GLMMadaptive package that uses adaptive Gaussian quadrature[64]. This approach allowed us to account for the data structure (zero-inflated, right-skewed continuous data, GLMMzi) and assess the relationship between density indices of the four large carnivore species and the set of explanatory variables described in Supplementary Table 9. All models were implemented in R using the packages GLMMadaptive and parallel. To account for potential problems of pseudo-replication, unit identity number was kept consistently as a random effect in all models for each of the 4 large carnivore species. Zero-inflated structures were added on all the fixed effects included in the models.

We reduced the full list of variables based on co-linearity and biological relevance to produce a set of 8 variables (e.g., collinearity between distance to closest roads and distance to closest settlements was too high and we chose to remove distance to closest roads from all the models). These 8 covariates included the 6 matching covariates described earlier (percentage of forest cover, terrain ruggedness, distance to closest settlements, human population density, latitude and longitude), to which we added year and protection status of the wildlife unit (protected or not, coded 0/1). Three interaction terms between fixed effects were

also added (between PA and Latitude, PA and Longitude and PA and years) to assess spatial and temporal variations in PA effectiveness.

A set of 12 models was built for each species. All 12 models included the 6 matching covariates and protection status at the unit scale (PA), therefore the difference in model structure resided in the addition of year, the three interactive terms and their different possible combinations (structure of the 12 models is described in the Supplementary Methods). All first-order model fits were ranked using the Akaike Information Criterion, the best model having the lowest AIC values from the set of 12 models built for each species.

The fixed effects estimates in mixed models with nonlinear link functions have an interpretation conditional on the random effects. To take this into account, we extracted the marginalized coefficients and their standard errors from the two part mixed models following the approach described by Hedeker et al.[65] and implemented by Rizopoulos[64] in the GLMMadaptive package.

Goodness-of-fit of the models was assessed using residual diagnostics following the procedures described in the DHARMa package. All statistical analyses were performed using the software R 3. 5. 1[60].

In order to test if PA size could affect our results, we built four additional models following the same procedure highlighted above. Data were restricted to protected wildlife units, density of the four species of carnivores was the response variables and covariates included: percentage of forest cover, human population density, distance to the closest settlements, terrain ruggedness, latitude, longitude, year and PA size. We extracted the marginalized coefficients and checked the residuals of the different models as highlighted above.

**Reporting summary**. Further information on research design is available in the Nature Research Reporting Summary linked to this article.

## Data availability
Data that support the findings of this study have been deposited in a Dryad repository, https://doi.org/10.5061/dryad.s1rn8pk4w. Spatial data from the WDPA World Database on Protected Areas are freely available at www.protectedplanet.net. Links for all the freely available predictor data sets are available in Supplementary Table 9.

## Code availability
R code that support the findings of this study has been deposited in a Dryad repository, https://doi.org/10.5061/dryad.s1rn8pk4w.

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

## Acknowledgements

Funding was provided by Maj ja Tor Nesslingin Säätiö (JT, Grant No. 201700201). We thank Prof. Ritzopoulos for helpful suggestions on the use of hurdle-mixed models and Dr. Barbara Class for help with statistical analyses.

## Author contributions

J.T. and M.C. designed the study. P.K. provided the FWTS dataset and expertize on the survey design and methodology. J.V.D. extracted environmental variables and provided expertize on spatial analysis. J.T. performed the effectiveness analyses. J.T. drafted the manuscript. All authors offered revision suggestions and approved the final version of the manuscript.

## Competing interests

The authors declare no competing interests.
