## [Peer Review File · Nature Communications]

Reviewers' comments first round:

Reviewer #1 (Remarks to the Author):

The paper provides a robust assessment of whether protected areas in Finland have helped conserve four large carnivore species. The study assesses a large (>2000 sites) database of PAs and non-PAs over a long time period (~30 years) for priority species (large carnivores), and contextualizes the findings in light of world policy conservation targets. The topic is relevant and the methods seem appropriate. In general, the study is noteworthy and could make a big splash in this time when measurement and evaluation of conservation approach is of utmost importance and awareness.

However, the narration and clarity of the main text need to be strengthened so that the results are explained as they are presented to make better use of the Nature Comms format. There are some minor issues with commas and grammar. Furthermore, the study needs to present statistical test values in the main text, not just the supplemental materials. My comments below attempt to provide suggestions but I encourage the authors to spend some time dwelling on the broader issues (like organization of the Results/Discussion).

COMMENTS:

The use of commas is inappropriate in some places and should be carefully reviewed throughout the paper. For example, in the Abstract commas should be removed in the following phrase (shown as suggested): "Here we used for the first time a combination of two methods...". And a comma should be inserted (shown as suggested with comma after 'governance'): "Our analysis highlights that PAs can fail to reach desired ecological outcomes, even in Global North nations with high national governance, and reinforce the combined importance...". In the Intro, comma should be inserted so the phrase reads: "Large carnivores are globally vulnerable to increasing anthropogenic pressures, with the populations..." On page 10, comma should be added so it reads: "The innovative approach presented here, combining both matching analyses and two part mixed models to estimate PA effectiveness, is particularly robust and complementary." Many other cases like this exist, where commas should be either removed or added.

Abstract:

- Change 'effects' to 'effect' in "...and a negative effects of PAs on lynx densities in Lapland"
- Remove 'Global North' to acknowledge with greater equality that nations across the world experience challenges. Phrase would then read: "...even in nations with high national governance...". Consider changing "high national governance" to "strong national governance".

Introduction:

- 1st paragraph: change "safeguarding biodiversity in front of increasing human pressures" to "safeguarding biodiversity against increasing human pressures"
- 1st par: structure of phrasing and thus meaning is not entirely clear in this phrase: "with first attempts to measure effectiveness focussing on gap analyses -measuring representativeness in terms of species diversity covered". Please reword for clarity.
- 1st par: This is optional since the intro is well organized and complete (although more concise than some readers may like), and I'll default to the Editor's opinion and space limitations, but it could be helpful to include another few sentences in the intro to more fully convey the predominant role of PAs in safeguarding biodiversity. They are the first and foremost tool for protecting species as well as boosting local community's livelihoods through ecotourism. Which makes it all the more shocking that their effectiveness hasn't been sufficiently explored. This could help readers who are less familiar with the PA literature better understand the motivation and significant of the current study.

- 1st par: Related to above comments, also important to acknowledge somewhere that carnivores do NOT represent all species. Although carnivores do represent indicator species given their important role in regulating ecosystem function, and are species highly valued by 'the public' (especially the public in Western countries, whose support drives and funds conservation globally especially for carnivores), the authors should take great care not to overstate their results as applicable to all species.

- 2nd par: Matching designs are not the only sampling design for testing the effectiveness of PAs, and the authors should acknowledge and cite in the intro some of the studies that have evaluated the effectiveness of PAs for protecting wildlife species. The current wording makes it sound like such studies have never been done. For example (the authors can choose whether to cite or not): Lindsey, P.A., Chapron, G., Petracca, L.S., Burnham, D., Hayward, M.W., Henschel, P., Hinks, E., Garnett, S.T., Macdonald, D.W., Macdonald, E.A., Ripple, W.J., Zander, K. & Dickman, A.J. (2017). Relative efforts of countries to conserve world's megafauna. *Glob. Ecol. Conserv.*, 10, 243–252.

3rd par: Here is an opportunity to explicitly state the goal of the study. At the end of this paragraph, state what the goal(s) of the study are (generally). Alternatively, could do this in the final few sentences of the Intro (after hypotheses, if included). Clearly and generally state the multiple methodology approaches and why each was used (e.g., First, to examine X, we did Y. Then, to explore W, we did Z.). Also clearly state which four carnivore species are examined, to set up the Results/Discussion section.

4th par: Parentheses needed around species scientific names. Greater precision and grammatical care is needed in the phrasing throughout. For example, change "Degradation and loss of habitat and human..." to "The degradation and loss of habitat in addition to human...". Change "have driven an historical reduction..." to "have historically driven a reduction...". May wish to edit to something like "...a reduction in most large carnivores globally" to convey the massive scale of decline.

4th par: Why is understanding outcomes "of the utmost importance"? This would be a good place to add in the ecological significance and human value of large carnivores. The last sentence is also a run on. Suggested edit: "Understanding the outcomes of conservation actions for this group of species is of the utmost importance, given the important ecological role of carnivores in regulating ecosystems and the profound significance of these species to people worldwide. Evaluating conservation actions is especially important in the European context, where large carnivore populations are recovering at a continental scale, although with important regional differences." Need to clarify what is meant by "important regional differences", since it is not clear in the 4th or even 5th par.

5th par: Clarify phrasing and precision. Suggested edit: "...through reducing direct killing of carnivores by humans and increasing the availability of resources, the latter mediated by the protection of habitat and prey populations." With regards to the "In Finland..." sentence about growing carnivore populations, need to tie this back more explicitly to PAs or hunting.

5th par: Make new paragraph at "On the other hand...". In the first paragraph, write a topical sentence that introduces the drivers you will discuss, as was done in the 4th par, such as "On the other hand, we expect PAs to have a negative effect on carnivores through...". The hypothesis that PAs will play different roles is not particularly informative or helpful for the reader in understanding concrete predictions (e.g., driver X results in positive PA outcomes for predators...). If the authors wish to include hypotheses (not necessary since the point presented in the intro is to evaluate effectiveness rather than identify drivers), please concretely present predicted drivers and how they tie to the metrics being assessed (carnivore population and/or densities).

Results/Discussion:

The narrative explanation around results currently does not enable the reader to easily take away the major conclusions or the reasons why results may have occurred. The narrative flow needs to be strengthened; below I offer some suggestions, but the authors should take considerable effort to strengthen the story behind their results. This is the advantage of having a section that

combines Results and Discussion, and the authors should take advantage of this format to explain their results as they present them. The organization of the results (absolute and temporal trends, followed by region) is fine, since the same drivers may affect species similarly.

The current text does not report any statistical test values in the main text, including t-tests and mixed effect models. Please report the values of the statistical tests when you refer to them in the main text. If there are too many tests to report, include the range of results to showcase the evidence for your statements.

1st par: change "susceptible" to "considered likely". Good to see Table S8 that explains what characteristics were used in the matching process.

1st par: change "Out of this" to "Out of these"

1st par: If the goal of the study is to assess carnivore populations using the Finish Wildlife Triangle Scheme data, why would you start by pairing PAs that don't include wildlife data? Why not narrow it down first to the PAs for which data are available?

Supp Tables are referring to "paired triangles" while the main text is referring to "paired units". Make consistent. Units may be better than triangles since it is not clear in the main text what a triangle is, and the definition of triangle in the methods could be counter-intuitive to some readers.

2nd par: The results need to more clearly be presented, both in terms of general wording of relationships and in terms of which results come from which pooling of data. For the first sentence, if these results are for data "when all years and all regions were pooled", start with this phrase to organize the presentation of results by the combination of data. Split up the first sentence into two or three sentences to walk the reader slowly through results without rushing. These are simple metrics but the wording obscures the meaning for the average reader, so explain simply in plain English what the relationship means. For example, instead of "there was a positive absolute PA effect on wolverine densities", write something like "Wolverine densities were higher in PAs than non-PAs".

Remove "significant" when presenting results; if you are discussing it, it should be statistically significant. Clarify "However, differences in individual years appear also as significant" to something like "Differences between species densities in PAs and non-PAs occurred in some years" (and explain why those years).

Given that the Results and Discussion are combined, it would be nice to explain the results as you present them so the study is in a more narrative and logical flow. "The PA network had a significant positive absolute effect on wolf, bear and wolverine densities only for one year out of the 29 considered in this study. Wolf densities were higher inside PAs than in unprotected sites in 2004, while both bear and wolverine densities were higher inside PAs than outside in 2005...". How fascinating – why might this have occurred?

"Yet the frequency of negative absolute effects of PAs on wolverine tended to increase over the study period (Supplementary Table S4)." Did you mean to also refer to Fig 2? In Fig. 2, both positive and negative effects increase over time; take care not to show bias in how results are being reported (or substantiate why a certain result is being highlighted when another is also present).

5th par: "...yet they seem to support higher densities of both lynx and bear." Higher than what?

5th par: ..." and could be linked to the distribution and impacts of small privately owned PAs." Why? Metapopulation dynamics? Connectivity? Consider including a few citations to support this hypothesis. And what else could be driving it? Though the paper is framed about PAs, other drivers could be at play and should be recognized.

6th par: "...it is worth highlighting negative, albeit non-significant, mean absolute PA effect on wolf densities in southern Finland..." Why is it worth highlighting? Explain the implications.

Pg 9: "a worrying trend" and "highlights worrying patterns". Reword to be non-emotional. Could use "alarming". Reword "at first" to refer to the type of analysis rather than the order in which it was done.

Pg. 10: Unpack the first paragraph and describe drivers in more detail.

Pg. 11: 3rd paragraph, include citations.

Pg. 12: "In summary, PAs in Finland do little for large carnivore populations" This is a BIG statement and generality that could have serious negative ramifications for PAs far outside of Finland. One could argue that your results show positive effects of many PAs, in addition to negative effects. If those PAs were not present, where would the species be? I am aware that these results are counter to what most conservationists are hoping you'll find, and you do need to accurately convey the conclusions of your results. However, statement like these can easily be picked up and mis-represented by the press, so be thoughtful in the concluding statements. Please refine this conclusion statement to include the nuance that is present in your results.

Pg. 12: Please rephrase this paragraph to suggest some solutions that could improve wildlife outcomes in PAs. As written ("We suggest...") it sounds as though these are results from your analyses.

Pg. 12: It's not clear how the beginning sentence introduces the rest of the paragraph. "Finally, we acknowledge that the species considered in this study are highly controversial and prone to trigger conflicts with local communities worldwide. Yet..." In fact BECAUSE they are "controversial" (and I would choose a different word, since this word as used prioritizes social values explicitly as a reason to or not to conserve), these species are in many ways ideal indicators of the utility of PAs in conserving far-roaming species that may rely on PAs as their core home but also could range beyond.

Pg. 12: "Yet these results prompt the question of what do protected areas aim to conserve in Global North countries." I do not agree with the juxtaposition of Global North vs. Global South used in this paper. It perpetuates excluding stereotypes about which hemisphere conserves "better", and does nothing to recognize or stimulate conversations about colonial conservation or the challenges faced by different countries across the world. Furthermore, Finland does not represent all countries in the northern hemisphere, nor does the article attempt to explain its representativeness. Unless the authors wish to more explicitly explain and centralize the framing in this light, please remove all mention of Global North or South because this framing limits rather than advances the conclusions and implications of the study.

Methods:

Did the authors explore the impact of PA size in the model? It strikes me that PAs could be meaningless for species with large home ranges like large carnivores if PAs are not sufficient enough in size to play a large role in their conservation.

Pg. 13: Did the authors verify the accurateness of the World Database of Protected Areas for Finnish PAs? They are often not accurate for African PAs.

Methods are generally very clearly written. Thank you for including QQ plots in supp.

The methodology and statistical analysis appear robust and appropriate. The dataset is impressive, and the methods appear to utilize the data well to answer the question at hand.

Although outside the scope of the questions raised in this paper, I would suggest that the authors consider repeating the study to answer questions addressing what makes a PA effective, including: the budget of the PA, the number of guards, the type of co-management, the amount of tourism (visitation or income), the human population surrounding the PA, and other such management/social variables which likely feed into to a more robust explanation of the findings in

the current study.

Figures:

- Fig 1 needs an inset showing where in the world the map is located. This map is very detailed so I would encourage the journal to publish it as large as possible. Would it be feasible to include a second map of forest cover, to support the statement in Results and Discussion sentence 1.

- Fig. 2, 4: As long as it complies with the journal requirements, I'd recommend adding either a vector shape or name of the species featured in each panel so the reader doesn't need to rely on the caption to know which panel represents which species.

Fig 2: In caption mention that asterisk indicates statistically significant difference between PA and non-PA. Watch capitalization of species name (use lowercase).

Fig 3: What do the two asterisk side-by-side indicate? Label that asterisk indicate statistical difference btw PAs and non-PAs. Include the sample size of matched PAs in each region on the graph or in the caption.

Table S1: Include SD or range for average PA size. The Game Management Area names need to be shown on a map somewhere so we can connect Fig 1 with Table S1.

Fig 4. The authors could consider flipping the axes of Fig 4 to match Fig 2, where up = positive and down = negative.

Fig 5. Change color scheme so it doesn't match Fig 4 (red/blue has different meaning).

References are not consistent in formatting.

Reviewer #2 (Remarks to the Author):

The authors use a dataset of impressive spatial and temporal extent to examine whether large carnivores tend to have higher abundances in protected areas than outside. Some important results are presented here, including the apparently novel finding that wolverines are declining within high-latitude PAs. However, I am concerned that not all of the authors' conclusions are well supported by the data and that the main message regarding a lack of effectiveness of protected areas is overstated.

In general, I'm not convinced that the conclusion stated in the title and repeated throughout the manuscript that "protected areas play little role in maintaining large carnivore populations" is correct. It seems that the real answer to the question of how PAs affect Finnish large carnivores is "it's complicated" (which admittedly is not nearly as catchy a title). According to the matching analysis results, both lynx and wolverine appear to benefit overall, despite wolverine declines in reindeer herding areas of Lapland. Bears benefit in the south and wolves may or may not have had higher abundance in PAs in at least some years. So, statements like "It is striking that our results indicate negative impacts of PAs on certain species of large carnivores..." (Discussion) seem poorly justified and could easily be misused by anyone interested in reducing protected area coverage.

It's also not clear to me that the matching analysis, which the authors note has previously been used for tree coverage to detect whether deforestation rates are higher outside of PAs, is necessarily appropriate for large carnivores like wolves and bears with often very large home ranges. If a network of smaller protected areas like those in the south of Finland are helping to maintain a wide-ranging population of wolves, would you expect to detect that effect by calculating abundance difference between single transects? In an area of low-density development mixed with small PAs, one could imagine that large carnivores are regularly moving between the PAs and the surrounding private/unprotected lands such that there would be no obvious difference in

abundance between these two categories. But that certainly does not mean that those protected areas and the connectivity between them are unimportant for large carnivores.

There are also some potential statistical issues with the matching analysis results. In Fig 2 and Table S4, the t-tests used to detect significant differences from zero here don't seem particularly trustworthy. The discrepancy between the t-test results and the 95% CIs for some years/species is notable and, given the sheer number of tests conducted (4 species x 30 yrs each) it seems the possibility of false positives is quite high. Indeed, with this number of tests and an alpha level of 0.05, you'd expect six significant values by chance alone (the authors report 7 significant values).

Results, referring to Fig 2: "...the frequency of negative absolute effects of PAs on wolverine tended to increase over the study period". Given the huge confidence intervals and the effect size hovering near zero, I'm not confident this statement is well supported by the data. Couldn't these dips below zero just be noise? And it seems to me there are similar number of positive values (also with huge variance) that are not similarly interpreted by the authors.

For the matching analysis, if I understand correctly, the authors compared the distribution of covariate values for PA triangles to the distribution of covariate values for non-PA matches and found that these distributions were similar enough to conclude that the overall matching exercise was successful. But is there any way to assess the quality of individual matches? Comparing the distributions of all covariate values in the PA or non-PA set doesn't really tell you how well any individual pair was matched, which would seem important since carnivore abundances are directly compared between matched pairs.

Methods: I find this statement confusing – "Following Rizopoulos⁴⁶, we used an ANOVA to compare the different fitted mixed models using a likelihood ratio test. All first-order models were ranked using the Akaike Information Criterion, the best model having the lowest AIC from the set of 12 models built for each species." This seems to say that both ANOVA and AIC were used to determine the best model. These are philosophically rather different methods, so it should be made clear which method actually resulted in the models being described in the results.

Minor Points:

Intro – It's not clear why the authors keep making the Global North vs. Global South distinction. If the point is that Finland has relatively good governance when it comes to managing wildlife/natural resources, this could simply be stated.

Last sentence of Intro: "...we hypothesize that PAs will play different role for the different carnivore species in different regions in Finland." This is quite vague and essentially has to be true. I would suggest presenting some specific hypotheses or just omitting this part.

Methods – Wildlife population time series data: The methods for conducting winter wildlife triangle counts are described in detail, but any differences for summer counts (which contributed all of the bear data) are not mentioned.

Methods: "A wildlife triangle is a permanent line transect route of 12 km (4 x 3 km)." I believe Intro says 10 km

Table S8: The authors refer to the metric calculated as the SD of elevation as "slope". Slope is typically calculated as the first derivative of elevation, while sd(elevation) is perhaps better thought of as a simple estimate of terrain ruggedness. It looks like the authors are ultimately interpreting this metric as ruggedness anyway, so I would just drop any mention of slope to avoid confusion.

Reviewer #3 (Remarks to the Author):

The paper "Protected areas play little role in maintaining large carnivore populations in Finland" is a simple analysis of a case study focusing on protected areas in Finland. I find the paper too narrow and not based on a proper understanding of its case study to be suitable for Nature Communications. The paper should however be publishable after revision in a more specialized journal, for example Wildlife Biology.

The first issue I have is that the authors assume by default that large carnivores are supposed to benefit from protected areas. I do not think this is supported by evidence. The impressive recovery that large carnivores have made in Europe has nothing to do with protected areas. Wolves for examples have not recovered in Poland, Italy, Slovenia, Sweden or Spain because of protected areas. They have recovered because of strict legal protection, abundant prey base and a public opinion generally supportive of nature conservation.

Large carnivores, especially in Northern Europe, have huge home ranges and I suspect that most of the protected areas are simply too small to host even a single wolf pack. Most individual wolves will move across a landscape that is un-protected simply because protected areas are too small. Therefore, I don't understand the logic behind assuming a role of protected areas. At least, information on the size of protected areas needs to be included in the analysis.

Next, even if protected by law, large carnivores are all hunted in Finland. I am not knowledgeable enough to tell whether they can also be hunted inside some or all protected areas but some European countries allow hunting in protected areas so this needs to be clarified. The paper ignores totally the fact that wolves are under Annex V of the Habitats Directive in the northern part of Finland, which Finland interprets as an exclusion zone. This certainly influences the results of the analysis (in one way or another).

The data used for the analysis need to be understood in a broader context. As the paper correctly reports, the triangles are surveyed by local hunters. This implies that the track counts may not be only track counts but what hunters want to let authorities believe. The problem here is that a stakeholder that is a party in the conflict with large carnivores is also responsible for data production. It would be naive to believe that this data production is not used for political purpose. In other words, the data are the product of a biological process and a political process and the relative shares of the two processes is unknown. A decline of tracks in a given part of the country may simply be caused by a passive protest by hunters against authorities.

The models need to be better presented. I am not familiar with the matching methods but the description needs to show the models (in equations). Same comments for the mixed effect models where a table showing model selection is needed.

Finally, the conclusions are not supported by the analysis. For example the authors recommend to improve management (but they have not identified where management fails and, frankly, which results would have led the authors to not argue for better management? If the conclusion is a generality independent from the analysis, it is more a personal opinion) and also creating anti-poaching units (but the authors have not quantified poaching – factors such as PA size, prey base or legal hunting are ignored).

We would like to thank the three reviewers for providing thoughtful comments on the last version of this manuscript. We have outlined below in detail how we have addressed each one of the reviewers' comments.

As suggested by the different reviewers and the Associate Editor:

- We have modified the structure of the Introduction and the 'Results and Discussion' sections in order to improve the flow and the understanding of our main results (as suggested by reviewer 1). We have also modified the main text and the title in order not to overstate our conclusions (suggested by reviewers 2 and 3); we also paid special attention to grammar throughout the MS;

- Figure 1 has been expanded and Figure 3 removed;

- Furthermore, we have recomputed the absolute PA effect to account for differences in the number of repeated observations amongst pairs of matched triangles. Therefore, for each species, we performed iterative random sampling (1000 iterations), selecting only one value of absolute PA effect for each pair of matched triangles before estimating the median PA effect across triangles. We have also adjusted p-values for multiple comparisons using the conservative Bonferroni correction as suggested by reviewer 2. This computation has resulted in slight change of results, and has eliminated some of the spurious regional, temporal or interspecific patterns that were previously more difficult to explain. The main message, that PAs have no effect on carnivores, remains, in these more robust analyses.

- Finally, the three reviewers and the Associate Editor were concerned about the potential effect of PA size on our results. Therefore, we tested the effect of PA size on carnivore densities by building new hurdle mixed-models adding this covariate to other environmental covariates previously included in the models. PA size had no effect on the densities of the 4 carnivores studied here and did not modify the effect of protection status (PA/no PA) on carnivore densities. We have added a paragraph in the Results section presenting these results.

We believe that these modifications have improved considerably the manuscript.

Dr. Julien Terraube

University of Helsinki, Finland

Reviewers' comments:

Reviewer #1 (Remarks to the Author):

The paper provides a robust assessment of whether protected areas in Finland have helped conserve four large carnivore species. The study assesses a large (>2000 sites) database of PAs and non-PAs over a long time period (~30 years) for priority species (large carnivores), and contextualizes the findings in light of world policy conservation targets. The topic is relevant and the methods seem appropriate. In general, the study is noteworthy and could make a big splash in this time when measurement and evaluation of conservation approach is of utmost importance and awareness.

However, the narration and clarity of the main text need to be strengthened so that the results are explained as they are presented to make better use of the Nature Comms format. There are some minor issues with commas and grammar. Furthermore, the study needs to present statistical test values in the main text, not just the supplemental materials. My comments below attempt to provide suggestions but I encourage the authors to spend some time dwelling on the broader issues (like organization of the Results/Discussion).

→ We would like to thank the reviewer for the thoughtful comments provided below.

COMMENTS:

The use of commas is inappropriate in some places and should be carefully reviewed throughout the paper. For example, in the Abstract commas should be removed in the following phrase (shown as suggested): "Here we used for the first time a combination of two methods...". And a comma should be inserted (shown as suggested with comma after 'governance'): "Our analysis highlights that PAs can fail to reach desired ecological outcomes, even in Global North nations with high national governance, and reinforce the combined importance...". In the Intro, comma should be inserted so the phrase reads: "Large carnivores are globally vulnerable to increasing anthropogenic pressures, with the populations..." On page 10, comma should be added so it reads: "The innovative approach presented here, combining both matching analyses and two part mixed models to estimate PA effectiveness, is particularly robust and complementary." Many other cases like this exist, where commas should be either removed or added.

→ We thank the reviewer for this comment. The sentences highlighted above by the reviewer have been changed as suggested. We have also carefully reviewed the use of commas throughout the MS.

Abstract:

- Change 'effects' to 'effect' in "...and a negative effects of PAs on lynx densities in Lapland"

→ The abstract has been rewritten entirely and this sentence has been removed.

Remove 'Global North' to acknowledge with greater equality that nations across the world experience challenges. Phrase would then read: "...even in nations with high national governance...". Consider changing "high national governance" to "strong national governance".

→ The abstract has been rewritten entirely and this sentence has also been removed.

Introduction:

- 1st paragraph: change “safeguarding biodiversity in front of increasing human pressures” to “safeguarding biodiversity against increasing human pressures”

→ We have changed this sentence as suggested by the reviewer.

- 1st par: structure of phrasing and thus meaning is not entirely clear in this phrase: “with first attempts to measure effectiveness focussing on gap analyses -measuring representativeness in terms of species diversity covered”. Please reword for clarity.

→ We have reworded this sentence in order to improve clarity as suggested by the reviewer (lines 75-76 p 3).

- 1st par: This is optional since the intro is well organized and complete (although more concise than some readers may like), and I’ll default to the Editor’s opinion and space limitations, but it could be helpful to include another few sentences in the intro to more fully convey the predominant role of PAs in safeguarding biodiversity. They are the first and foremost tool for protecting species as well as boosting local community’s livelihoods through ecotourism. Which makes it all the more shocking that their effectiveness hasn’t been sufficiently explored. This could help readers who are less familiar with the PA literature better understand the motivation and significance of the current study.

→ We have added a sentence developing the importance of PAs, including key references to convey the importance of a better understanding of these biodiversity conservation instruments (lines 71-75 p 3)

- 1st par: Related to above comments, also important to acknowledge somewhere that carnivores do NOT represent all species. Although carnivores do represent indicator species given their important role in regulating ecosystem function, and are species highly valued by ‘the public’ (especially the public in Western countries, whose support drives and funds conservation globally especially for carnivores), the authors should take great care not to overstate their results as applicable to all species.

→ We agree with the reviewer and we have added a short paragraph highlighting these points. See lines 116-119 p 4.

- 2nd par: Matching designs are not the only sampling design for testing the effectiveness of PAs, and the authors should acknowledge and cite in the intro some of the studies that have evaluated the effectiveness of PAs for protecting wildlife species. The current wording makes it sound like such studies have never been done. For example (the authors can choose whether to cite or not): Lindsey, P.A., Chapron, G., Petracca, L.S., Burnham, D., Hayward, M.W., Henschel, P., Hinks, E., Garnett, S.T., Macdonald, D.W., Macdonald, E.A., Ripple, W.J., Zander, K. & Dickman, A.J. (2017). Relative efforts of countries to conserve world’s megafauna. *Glob. Ecol. Conserv.*, 10, 243–252.

→ We agree with the reviewer, matching designs are not the only sampling design for testing the effectiveness of PAs in maintaining wildlife species. We have rewritten the introduction to clarify a few matters. We recognize that substantial work has addressed issues of coverage, whether habitat or range coverage by protected areas. We refer to these studies as focusing on efficiency, as opposite to effectiveness, and believe that the reference mentioned by the reviewer belongs to this

group of studies (see lines 89-93 p 3). Eventually, we want protected area networks that do cover the species and their habitats, but we want also PAs that are able to effectively reduce threats or manage populations. Counterfactual, and also other approaches, do focus on this second aspect. We hope that the text now better reflects such different approaches, but also that the title of the paper better clarifies the focus of our work.

3rd par: Here is an opportunity to explicitly state the goal of the study. At the end of this paragraph, state what the goal(s) of the study are (generally). Alternatively, could do this in the final few sentences of the Intro (after hypotheses, if included). Clearly and generally state the multiple methodology approaches and why each was used (e.g., First, to examine X, we did Y. Then, to explore W, we did Z.). Also clearly state which four carnivore species are examined, to set up the Results/Discussion section.

→ We have modified this paragraph and stated clearly what the main aim of our study is. See lines 99-106 p 3 and added predictions at the end of the Introduction (lines 133-141 p 4).

4th par: Parentheses needed around species scientific names. Greater precision and grammatical care is needed in the phrasing throughout. For example, change “Degradation and loss of habitat and human...” to “The degradation and loss of habitat in addition to human...”. Change “have driven an historical reduction...” to “have historically driven a reduction...”. May wish to edit to something like “...a reduction in most large carnivores globally” to convey the massive scale of decline.

→ We have added parentheses around species scientific names. We have also edited the three sentences highlighted above to implement the modifications suggested by the reviewer. See lines 108-112 p 4.

4th par: Why is understanding outcomes “of the utmost importance”? This would be a good place to add in the ecological significance and human value of large carnivores. The last sentence is also a run on. Suggested edit: “Understanding the outcomes of conservation actions for this group of species is of the utmost importance, given the essential ecological role of carnivores in regulating ecosystems and the profound significance of these species to people worldwide. Evaluating conservation actions is especially important in the European context, where large carnivore populations are recovering at a continental scale, although with important regional differences.” Need to clarify what is meant by “important regional differences”, since it is not clear in the 4th or even 5th par.

→ We agree with the reviewer on this point. We have edited the sentence following the suggestions mentioned above. See lines 112-114 p 4. We have deleted the part mentioning “important regional differences” as this was referring to local population declines of certain carnivore species despite the overall recovery, like for example the lynx in the Balkans. These aspects are not a central point to this study and we preferred to avoid any further confusions.

5th par: Clarify phrasing and precision. Suggested edit: “...through reducing direct killing of carnivores by humans and increasing the availability of resources, the latter mediated by the protection of habitat and prey populations.” With regards to the “In Finland...” sentence about growing carnivore populations, need to tie this back more explicitly to PAs or hunting.

→ This paragraph has been entirely rewritten. See lines 120-141 p 4. We have added a sentence mentioning the links between the recovery of large carnivores in Finland and hunting regulations. See lines 126-127 p 4.

5th par: Make new paragraph at “On the other hand...”. In the first paragraph, write a topical sentence that introduces the drivers you will discuss, as was done in the 4th par, such as “On the other hand, we expect PAs to have a negative effect on carnivores through...”. The hypothesis that PAs will play different roles is not particularly informative or helpful for the reader in understanding concrete predictions (e.g., driver X results in positive PA outcomes for predators...). If the authors wish to include hypotheses (not necessary since the point presented in the intro is to evaluate effectiveness rather than identify drivers), please concretely present predicted drivers and how they tie to the metrics being assessed (carnivore population and/or densities).

→ This paragraph has been entirely rewritten, therefore the sentences highlighted above by the reviewer do not appear anymore in the new version of the MS. See lines 121-141 p 4. We have also added 3 predictions about expected variations in PA effectiveness among species of carnivores, between Finnish regions and over the 30-year study period. See lines 136-141 p 4.

Results/Discussion:

The narrative explanation around results currently does not enable the reader to easily take away the major conclusions or the reasons why results may have occurred. The narrative flow needs to be strengthened; below I offer some suggestions, but the authors should take considerable effort to strengthen the story behind their results. This is the advantage of having a section that combines Results and Discussion, and the authors should take advantage of this format to explain their results as they present them. The organization of the results (absolute and temporal trends, followed by region) is fine, since the same drivers may affect species similarly.

→ We thank the reviewer for this comment. We have paid special attention to restructure this new version of the MS in order to strengthen the narrative flow.

The current text does not report any statistical test values in the main text, including t-tests and mixed effect models. Please report the values of the statistical tests when you refer to them in the main text. If there are too many tests to report, include the range of results to showcase the evidence for your statements.

→ We agree with the reviewer and we have now added statistical test values in the main text. See lines 165 p6-line 264 p 10.

1st par: change “susceptible” to “considered likely”.

→ We have done as suggested.

Good to see Table S8 that explains what characteristics were used in the matching process.

1st par: change “Out of this” to “Out of these”

→ We have done as suggested.

1st par: If the goal of the study is to assess carnivore populations using the Finish Wildlife Triangle Scheme data, why would you start by pairing PAs that don't include wildlife data? Why not narrow it down first to the PAs for which data are available?

→ I guess this is a misunderstanding. All triangles have wildlife data but triangles can have different combinations of species detected during winter or summer surveys. All triangles included in the matching analyses include at least density data of one carnivore species. We have reworded this sentence to avoid further confusions (see lines 149-151 p 4-5).

Supp Tables are referring to "paired triangles" while the main text is referring to "paired units". Make consistent. Units may be better than triangles since it is not clear in the main text what a triangle is, and the definition of triangle in the methods could be counter-intuitive to some readers.

→ Thank you for this comment. We have now replaced paired triangles by paired units throughout the text.

2nd par: The results need to more clearly be presented, both in terms of general wording of relationships and in terms of which results come from which pooling of data. For the first sentence, if these results are for data "when all years and all regions were pooled", start with this phrase to organize the presentation of results by the combination of data. Split up the first sentence into two or three sentences to walk the reader slowly through results without rushing. These are simple metrics but the wording obscures the meaning for the average reader, so explain simply in plain English what the relationship means. For example, instead of "there was a positive absolute PA effect on wolverine densities", write something like "Wolverine densities were higher in PAs than non-PAs".

→ We have followed the suggestions above in order to improve the presentation of our results.

Remove "significant" when presenting results; if you are discussing it, it should be statistically significant. Clarify "However, differences in individual years appear also as significant" to something like "Differences between species densities in PAs and non-PAs occurred in some years" (and explain why those years).

→ We have removed "significant" from the text in the Results and Discussion section. As you can see in the Supp. Table S4, we have adjusted p-values for multiple comparisons using the conservative Bonferroni correction. The average absolute protected area effect per year is never significantly different from 0 for any of the four species of large carnivores, therefore we chose not to mention interannual differences in the new version of this MS.

Given that the Results and Discussion are combined, it would be nice to explain the results as you present them so the study is in a more narrative and logical flow. "The PA network had a significant positive absolute effect on wolf, bear and wolverine densities only for one year out of the 29 considered in this study. Wolf densities were higher inside PAs than in unprotected sites in 2004, while both bear and wolverine densities were higher inside PAs than outside in 2005...". How fascinating – why might this have occurred?

→ We have restructured the whole Results/Discussion section in order to integrate more the discussion of results in the text as suggested by the reviewer (see lines 187-220 p 7 and lines 265-

297 p 10). For this particular suggestion, however, the results have changed after applying Bonferroni corrections (see comments to editor and answer above), and thus the paragraph has been deleted.

“Yet the frequency of negative absolute effects of PAs on wolverine tended to increase over the study period (Supplementary Table S4).” Did you mean to also refer to Fig 2? In Fig. 2, both positive and negative effects increase over time; take care not to show bias in how results are being reported (or substantiate why a certain result is being highlighted when another is also present).

→ We agree with the reviewer here. Therefore we have removed this sentence from the text. In addition, as the results have changed considerably after adjusting p-values with a Bonferroni correction, we state clearly that there are no clear significant annual differences in the absolute PA effect for any species (Lines 172-174 p 6).

5th par: “...yet they seem to support higher densities of both lynx and bear.” Higher than what?

→ This whole paragraph has been rewritten following changes in the results linked to regional variations in the absolute PA effect.

5th par: ...” and could be linked to the distribution and impacts of small privately owned PAs.” Why? Metapopulation dynamics? Connectivity? Consider including a few citations to support this hypothesis. And what else could be driving it? Though the paper is framed about PAs, other drivers could be at play and should be recognized.

→ We agree with the reviewer about the potential impact of other drivers on carnivore densities like connectivity between PAs. Yet the positive impacts of PAs on bear and lynx densities in Southern Finland we were trying to explain, were spurious patterns that have vanished with the most robust analyses applied now - the implementation of iterative random sampling to calculate the absolute PA effect. Thus there is no need to list the suggested potential drivers.

6th par: “...it is worth highlighting negative, albeit non-significant, mean absolute PA effect on wolf densities in southern Finland...” Why is it worth highlighting? Explain the implications.

→ Similarly, this result does not hold anymore in the newest version of the analyses. This sentence has thus been removed from the text.

Pg 9: “a worrying trend” and “highlights worrying patterns”. Reword to be non-emotional. Could use “alarming”. Reword “at first” to refer to the type of analysis rather than the order in which it was done.

→ We agree with the reviewer. We have been careful to use only non-emotional wording throughout the text of the MS.

Pg. 10: Unpack the first paragraph and describe drivers in more detail.

→ We have rewritten this paragraph and developed further the potential drivers explaining the difference in PA effects on bear densities obtained through the two methodologies (see lines 307-321 p 11).

Pg. 11: 3rd paragraph, include citations.

→ We have now included more citations to back up statements in this paragraph.

Pg. 12: “In summary, PAs in Finland do little for large carnivore populations” This is a BIG statement and generality that could have serious negative ramifications for PAs far outside of Finland. One could argue that your results show positive effects of many PAs, in addition to negative effects. If those PAs were not present, where would the species be? I am aware that these results are counter to what most conservationists are hoping you’ll find, and you do need to accurately convey the conclusions of your results. However, statement like these can easily be picked up and misrepresented by the press, so be thoughtful in the concluding statements. Please refine this conclusion statement to include the nuance that is present in your results.

→ We thank the reviewer for highlighting this. We note that carnivore trends in Finland are overall increasing (see the ‘year’ effect in the hurdle models, Supp. Info), as they are in Europe at large. Thus we cannot say that PAs would not have value. They indeed have value together with other non-protected habitat, but our results indicate that PAs do not support higher carnivore densities than non-protected sites, and thus their role for carnivores may be just complementary habitat.

We have reworded this sentence and toned-down in general our main message throughout the MS (the title has also been changed accordingly).

Pg. 12: Please rephrase this paragraph to suggest some solutions that could improve wildlife outcomes in PAs. As written (“We suggest...”) it sounds as though these are results from your analyses.

→ We entirely modified the structure of Discussion and this sentence was removed during the process.

Pg. 12: It’s not clear how the beginning sentence introduces the rest of the paragraph. “Finally, we acknowledge that the species considered in this study are highly controversial and prone to trigger conflicts with local communities worldwide. Yet...” In fact BECAUSE they are “controversial” (and I would choose a different word, since this word as used prioritizes social values explicitly as a reason to or not to conserve), these species are in many ways ideal indicators of the utility of PAs in conserving far-roaming species that may rely on PAs as their core home but also could range beyond.

→ We appreciate this comment and have rephrased this sentence accordingly. See lines 333-349 p 12.

Pg. 12: “Yet these results prompt the question of what do protected areas aim to conserve in Global North countries.” I do not agree with the juxtaposition of Global North vs. Global South used in this paper. It perpetuates excluding stereotypes about which hemisphere conserves “better”, and does nothing to recognize or stimulate conversations about colonial conservation or the challenges faced by different countries across the world. Furthermore, Finland does not represent all countries in the northern hemisphere, nor does the article attempt to explain its representativeness. Unless the authors wish to more explicitly explain and centralize the framing in this light, please remove all mention of Global North or South because this framing limits rather than advances the conclusions and implications of the study.

→ The aim of using Global North was definitely not to perpetuate excluding stereotypes or state that Global North countries are those conserving better than Global South countries. We initially, and perhaps mistakenly or naively used Global North to refer to “assumed stereotypes”, countries of generally good governance, and thought of having effective conservation. But we understand the misuse of the term, removed it and edited the sentence further - see line 333-349 p 12.

Methods:

Did the authors explore the impact of PA size in the model? It strikes me that PAs could be meaningless for species with large home ranges like large carnivores if PAs are not sufficient enough in size to play a large role in their conservation.

→ As suggested, we have tested the effect of PA size on carnivore densities by building new hurdle mixed-models adding this covariate to other environmental covariates previously included in the models. PA size had no effect on the densities of the 4 carnivores studied here and did not modify the effect of protection status (PA/no PA) on carnivore densities. We have added a paragraph in the Results section presenting these results (see lines 260-264 p10).

Pg. 13: Did the authors verify the accurateness of the World Database of Protected Areas for Finnish PAs? They are often not accurate for African PAs.

→ We confirm that we have verified the accurateness of the World Database of Protected Areas for Finnish PAs by checking the percentage of overlap between this source of delineation of the Finnish PAs and another one obtained from SYKE the Finnish Environment Institute. We did this for 10 PAs that were randomly selected and the percentage of overlap was always >90%.

Methods are generally very clearly written. Thank you for including QQ plots in supp. The methodology and statistical analysis appear robust and appropriate. The dataset is impressive, and the methods appear to utilize the data well to answer the question at hand.

→ We thank the reviewer for this positive feedback.

Although outside the scope of the questions raised in this paper, I would suggest that the authors consider repeating the study to answer questions addressing what makes a PA effective, including: the budget of the PA, the number of guards, the type of co-management, the amount of tourism (visitation or income), the human population surrounding the PA, and other such management/social variables which likely feed into to a more robust explanation of the findings in the current study.

→ We thank the reviewer for this suggestion. A further study including the suggested co-variables would be important. However we note that the analyses conducted here is an analysis at the PA-network level, answering the general question “Are Finnish PAs effective?”. The study does not look at effectiveness of individual PAs and variation between them. One of us has conducted a similar study for Madagascar PAS (Eklund et al 2019), yet we note that to apply matching at the individual PA level one needs multiple pairs of data per PA, which will not be available for population trends – but it is typically available for deforestation data. Nonetheless, one could address the question with a different modelling approach.

Figures:

- Fig 1 needs an inset showing where in the world the map is located. This map is very detailed so I would encourage the journal to publish it as large as possible. Would it be feasible to include a second map of forest cover, to support the statement in Results and Discussion sentence 1.

→ We have done as suggested. The new Figure 1 includes now 3 maps: the main one, similar to previous version, showing spatial variations in carnivore densities, and two smaller ones in the periphery showing the extent of forest cover in Finland and the location of Finland in the world.

- Fig. 2, 4: As long as it complies with the journal requirements, I'd recommend adding either a vector shape or name of the species featured in each panel so the reader doesn't need to rely on the caption to know which panel represents which species.

→ We have done as suggested and the new Figures 2 and 4 include the name of the four carnivore species included in this study.

Fig 2: In caption mention that asterisk indicates statistically significant difference between PA and non-PA. Watch capitalization of species name (use lowercase).

→ The new results presented in Figure 2 do not include any statistically significant annual absolute PA effect.

Fig 3: What do the two asterisk side-by-side indicate? Label that asterisk indicate statistical difference btw PAs and non-PAs. Include the sample size of matched PAs in each region on the graph or in the caption.

→ Figure 3 has now be removed from the new version of this MS, as none of the regional difference was significant. This result is now cited in the text only.

Table S1: Include SD or range for average PA size. The Game Management Area names need to be shown on a map somewhere so we can connect Fig 1 with Table S1.

→ We have included SD for average PA size in Table S1 as suggested. The Game Management Area names appear on the new version of Fig. 1.

Fig 4. The authors could consider flipping the axes of Fig 4 to match Fig 2, where up = positive and down = negative.

→ We have tried to flip the axes of Fig. 4 but the legend of all covariates switched to the x axis made the Figure difficult to interpret. Therefore, we chose keep the axes of Fig. 4 as they were.

Fig 5. Change color scheme so it doesn't match Fig 4 (red/blue has different meaning).

→ We have changed the colour scheme in Figure 4, therefore Fig. 4 and Fig. 5 do not have matching colour schemes in this new version.

References are not consistent in formatting.

→ We have revised the reference format and it should be consistent in the newest version of this MS.

Reviewer #2 (Remarks to the Author):

The authors use a dataset of impressive spatial and temporal extent to examine whether large carnivores tend to have higher abundances in protected areas than outside. Some important results are presented here, including the apparently novel finding that wolverines are declining within high-latitude PAs. However, I am concerned that not all of the authors' conclusions are well supported by the data and that the main message regarding a lack of effectiveness of protected areas is overstated.

In general, I'm not convinced that the conclusion stated in the title and repeated throughout the manuscript that "protected areas play little role in maintaining large carnivore populations" is correct. It seems that the real answer to the question of how PAs affect Finnish large carnivores is "it's complicated" (which admittedly is not nearly as catchy a title). According to the matching analysis results, both lynx and wolverine appear to benefit overall, despite wolverine declines in reindeer herding areas of Lapland. Bears benefit in the south and wolves may or may not have had higher abundance in PAs in at least some years. So, statements like "It is striking that our results indicate negative impacts of PAs on certain species of large carnivores..." (Discussion) seem poorly justified and could easily be misused by anyone interested in reducing protected area coverage.

→ We agree with the reviewer that the title was an overstatement of the results presented in the last version, and that the choice of wording may have been misleading. We agree that the role of PAs is complicated, and that they certainly have a complementary value, and obviously their role would be crucial if there would be no other suitable habitat outside the PAs. Furthermore, some of the results we previously presented, with regional or interspecific differences not easy to explain, have now changed after improving the robustness of the analyses. The results throughout are now more consistent, retaining the message that there are no differences between carnivore abundances inside and outside Finnish PAs. We further discuss whether such differences should be expected, as well as provide a broader view of the role of PAs. We thus have changed the title in this new version and toned down the text regarding the interpretation of the overall impacts of PAs on large carnivore populations throughout the MS.

It's also not clear to me that the matching analysis, which the authors note has previously been used for tree coverage to detect whether deforestation rates are higher outside of PAs, is necessarily appropriate for large carnivores like wolves and bears with often very large home ranges. If a network of smaller protected areas like those in the south of Finland are helping to maintain a wide-ranging population of wolves, would you expect to detect that effect by calculating abundance difference between single transects? In an area of low-density development mixed with small PAs, one could imagine that large carnivores are regularly moving between the PAs and the surrounding private/unprotected lands such that there would be no obvious difference in abundance between these two categories. But that certainly does not mean that those protected areas and the connectivity between them are unimportant for large carnivores.

→ We partially agree with the reviewer's comment. Indeed, this is probably what we see in our results: that PAs and other non-PA habitat complement each other supporting some larger home-ranges for carnivore species. However, we would expect that carefully chosen PAs, despite being

small, are protecting prime habitat with higher resources, less disturbance, etc and thus we would expect more frequent use of PAs than non-PAs. As for the large PA areas in the north, which should be able to contain the whole home range of carnivore species, we should expect larger differences with non-PA habitat. We agree with the reviewer that the impacts of PAs on large carnivore populations are more complicated to grasp and there are limits to our approach: for example the results for the wolf, lynx and wolverine correspond to winter densities, and do not allow to infer the effects of PAs on these species in summer. And one could actually expect seasonal differences in the effects of PAs on these species with higher reliance on protected areas during the denning season, i.e. spring/summer than in winter when carnivores might be more dependent on anthropogenic landscapes harbouring higher prey availability. Further research should decipher the importance of PAs for these species in other seasons and at finer scale (using GPS tagging for example). We have added these limits of our data in the main text (see lines 269-279 p 10).

There are also some potential statistical issues with the matching analysis results. In Fig 2 and Table S4, the t-tests used to detect significant differences from zero here don't seem particularly trustworthy. The discrepancy between the t-test results and the 95% CIs for some years/species is notable and, given the sheer number of tests conducted (4 species x 30 yrs each) it seems the possibility of false positives is quite high. Indeed, with this number of tests and an alpha level of 0.05, you'd expect six significant values by chance alone (the authors report 7 significant values).

→ We thank the reviewer for this note, and we fully agree. We have now adjusted the p-values with the Bonferroni correction and none of the annual absolute PA effect is statistically different from 0. See text in the new results section (lines 172-174 p 6 and Supp. Table S4).

Results, referring to Fig 2: "...the frequency of negative absolute effects of PAs on wolverine tended to increase over the study period". Given the huge confidence intervals and the effect size hovering near zero, I'm not confident this statement is well supported by the data. Couldn't these dips below zero just be noise? And it seems to me there are similar number of positive values (also with huge variance) that are not similarly interpreted by the authors.

→ We agree with the reviewer here. We have now chosen not to discuss in the MS the interannual variations in the absolute PA effect found for each species are not discussed, given the confidence intervals and the effect size close to 0.

For the matching analysis, if I understand correctly, the authors compared the distribution of covariate values for PA triangles to the distribution of covariate values for non-PA matches and found that these distributions were similar enough to conclude that the overall matching exercise was successful. But is there any way to assess the quality of individual matches? Comparing the distributions of all covariate values in the PA or non-PA set doesn't really tell you how well any individual pair was matched, which would seem important since carnivore abundances are directly compared between matched pairs.

→ We understand the reviewer's concern regarding the point mentioned above. Diagnosing the quality of the resulting matched samples is an essential step in the matching process. This is based on the assessment of the covariate balance in the matched groups, where balance is defined as the similarity of the empirical distributions of the full set of covariates in the matched treated and

control groups. There is no way of assessing the quality of individual matches. See Stuart, 2010 for further details.

Stuart EA. Matching methods for causal inference: a review and a look forward. *Statistical Science* 2010; 25(1): 1– 21.

Methods: I find this statement confusing – “Following Rizopoulos⁴⁶, we used an ANOVA to compare the different fitted mixed models using a likelihood ratio test. All first-order models were ranked using the Akaike Information Criterion, the best model having the lowest AIC from the set of 12 models built for each species.” This seems to say that both ANOVA and AIC were used to determine the best model. These are philosophically rather different methods, so it should be made clear which method actually resulted in the models being described in the results.

→ We agree with the reviewer that the phrasing of this paragraph was unclear. Model fits were ranked according to their AIC values. We have rewritten this part of the Methods (lines 457-459 p 14) and added a table with the AIC, BIC and log.Lik values of each of the 12 models built per species in the Supplementary Material (see Table S10).

Minor Points:

Intro – It’s not clear why the authors keep making the Global North vs. Global South distinction. If the point is that Finland has relatively good governance when it comes to managing wildlife/natural resources, this could simply be stated.

→ We agree with the reviewer and we have removed all references to Global North vs. Global South distinction.

Last sentence of Intro: “...we hypothesize that PAs will play different role for the different carnivore species in different regions in Finland.” This is quite vague and essentially has to be true. I would suggest presenting some specific hypotheses or just omitting this part.

→ We agree with the reviewer about the need to restructure the end of the Introduction. We have now added specific hypotheses and predictions in the last paragraph of the Introduction. See lines 133-141 p 4.

Methods – Wildlife population time series data: The methods for conducting winter wildlife triangle counts are described in detail, but any differences for summer counts (which contributed all of the bear data) are not mentioned.

→ The surveys conducted on the wildlife triangles during summer and winter are performed in a similar way. The only difference is that surveys performed during the winter were snow track survey while surveys performed during the summer were mud track surveys.

Methods: “A wildlife triangle is a permanent line transect route of 12 km (4 x 3 km).” I believe Intro says 10 km

→ We thank the reviewer for noticing this mistake. We have now corrected the length of the transect routes in the Introduction (see line 101 p 3).

Table S8: The authors refer to the metric calculated as the SD of elevation as “slope”. Slope is typically calculated as the first derivative of elevation, while sd (elevation) is perhaps better thought of as a simple estimate of terrain ruggedness. It looks like the authors are ultimately interpreting this metric as ruggedness anyway, so I would just drop any mention of slope to avoid confusion.

→ We agree with the reviewer and we have changed “slope “to “terrain ruggedness “throughout the text.

Reviewer #3 (Remarks to the Author):

The paper "Protected areas play little role in maintaining large carnivore populations in Finland" is a simple analysis of a case study focusing on protected areas in Finland. I find the paper too narrow and not based on a proper understanding of its case study to be suitable for Nature Communications. The paper should however be publishable after revision in a more specialized journal, for example Wildlife Biology.

→ We thank the reviewer for this perspective. These comments indicate that we did not make clear enough the main novelty of the paper. We understand that if taken as a case study, the exercise could be seen as narrow in scope, and not worthy for this journal. However we stress and clarify better in the paper that the approach developed here is at the forefront of research on statistical approaches for assessments of protected area effectiveness. A robust evaluation of the ecological outcomes of PAs is a timely and global endeavor, when knowledge about PA effectiveness is still scarce and preparations for the post-2020 Biodiversity Framework are ongoing. Studies assessing the effectiveness of PAs in maintaining wildlife while taking into account potential confounding factors are extremely rare due to poor geographical coverage and lack of wildlife data outside PAs. This is to our knowledge the first attempt to perform such analysis under a strict counterfactual scenario, which has been highlighted as a priority to improve assessments of PA effectiveness by recent high/profile articles on the same topic, e.g. Barnes et al. 2016 and Gray et al. 2016. This has been possible because of the unique, extensive and comprehensive dataset that we have analyzed, i.e. density data collected through 2171 wildlife transects covering a whole country over 30 years. In addition we have combined two methodologies: matching analyses and hurdle mixed-analyses to refine the estimates of protected area effectiveness and understand further the spatial and temporal patterns observed at the national scale. While partially similar methodologies have been applied to raster data, typically forest data, before, this is the first application to wildlife population data, consequently bringing about new challenges and questions that we hope to open with the paper. We have edited the text to highlight this methodological and conceptual relevance, while also modifying the discussion to better account for particularities of the results and insights about the case in question (see the new conclusion for example).

The first issue I have is that the authors assume by default that large carnivores are supposed to benefit from protected areas. I do not think this is supported by evidence. The impressive recovery that large carnivores have made in Europe has nothing to do with protected areas. Wolves for examples have not recovered in Poland, Italy, Slovenia, Sweden or Spain because of protected areas. They have recovered because of strict legal protection, abundant prey base and a public opinion generally supportive of nature conservation.

→ We agree with the reviewer that positive trends of carnivores in Europe are a consequence of many socio-political changes, but they are also related to land cover changes that have favoured suitable habitat. Protected and 'non-protected but suitable' habitat complement each other in supporting carnivore ranges. Public opinion in Finland towards carnivores varies regionally and poaching is still a problem. Legal protection at national level plays an important role, but protected areas with different regulatory mandates, offer, at least in theory, protection against disturbance and potentially more abundant resources. Although we do not expect large differences in abundances within and outside protected areas, we do expect some, and so does the public and the tourism industry. As large carnivore species are prone to trigger important social conflicts, these species are thus in many ways ideal indicators of the utility of PAs in conserving far-roaming predator species that may rely on PAs as their core home but also could range beyond due their large home range sizes. We now have clarified these points in the manuscript and finished the introduction with specific hypotheses on the effects of PAs on large carnivore species (see lines 133-141 p 4).

Large carnivores, especially in Northern Europe, have huge home ranges and I suspect that most of the protected areas are simply too small to host even a single wolf pack. Most individual wolves will move across a landscape that is un-protected simply because protected areas are too small. Therefore, I don't understand the logic behind assuming a role of protected areas. At least, information on the size of protected areas needs to be included in the analysis.

We thank the reviewer for this point. It is worth noting – now clarified in the ms- that we do not assume populations of carnivores to be confined to PAs, and indeed some carnivore individuals may add counts both for the outside and inside triangles compared. But if PA habitat were to offer more abundant resources, resting or hiding locations, one should expect differences in the frequency of use of PAs vs not PAs, which would be reflected in the estimated abundances. However, in order to account for the reviewer's concerns we have now tested the effect of PA size on carnivore densities using the hurdle mixed models in addition to other confounding factors (same environmental variables described in the Method sections). PA size did not affect the density of any of the species. We added this information in the Results and Discussion section (see lines 260-264 p4).

Next, even if protected by law, large carnivores are all hunted in Finland. I am not knowledgeable enough to tell whether they can also be hunted inside some or all protected areas but some European countries allow hunting in protected areas so this needs to be clarified. The paper ignores totally the fact that wolves are under Annex V of the Habitats Directive in the northern part of Finland, which Finland interprets as an exclusion zone. This certainly influences the results of the analysis (in one way or another).

→ Large carnivore can indeed be hunted in Finland and a small number of derogations permits are granted every year to hunt large carnivores inside protected areas on grounds of prevention of damages to livestock farming for example. We have now added this information in the main text (lines 196-198 p 7). The wolf is indeed virtually absent from the reindeer herding area meaning that no permanent wolf pack can be encountered in this area due to high hunting pressure. This means that the wolf densities encountered in the Lappish triangles are extremely low. This is visible on the Table S5 from the Supplementary Material where the absolute PA effect for the wolf could not be calculated when restricting the analysis to Lapland.

The data used for the analysis need to be understood in a broader context. As the paper correctly reports, the triangles are surveyed by local hunters. This implies that the track counts may not be only track counts but what hunters want to let authorities believe. The problem here is that a stakeholder that is a party in the conflict with large carnivores is also responsible for data production. It would be naive to believe that this data production is not used for political purpose. In other words, the data are the product of a biological process and a political process and the relative shares of the two processes is unknown. A decline of tracks in a given part of the country may simply be caused by a passive protest by hunters against authorities.

→ We agree with the reviewer on this point. However, although there are potential biases in the data linked to the collection process, we expect biases to affect similarly the treatment (protected sites) and control groups (non-protected sites) and thus not to bias our estimates of PA effectiveness. Potential biases could affect regional comparisons due to differential stakeholder participation, and different regional attitudes. In general, as this data is being used nationally to estimate carnivore populations and set hunting quotas, the expected biases are towards higher abundances.

The models need to be better presented. I am not familiar with the matching methods but the description needs to show the models (in equations). Same comments for the mixed effect models where a table showing model selection is needed.

→ A table showing the AIC, BIC and log.Lik values of the 12×4 models built (4 species of carnivores) has now been added to the Supplementary Material, see Supplementary Table S10 p 15-16. As suggested, we have also added the equation of the matching model in the Supplementary Material. See legend of Fig. S1 p 12.

Finally, the conclusions are not supported by the analysis. For example the authors recommend to improve management (but they have not identified where management fails and, frankly, which results would have led the authors to not argue for better management? If the conclusion is a generality independent from the analysis, it is more a personal opinion) and also creating anti-poaching units (but the authors have not quantified poaching – factors such as PA size, prey base or legal hunting are ignored).

→ We have restructured the whole Results and Discussion section. We particularly tone-down the last three paragraphs to be more conservative about the conclusions we develop and the recommendations we highlight (lines 299-348 p 10-12).

Reference cited:

Barnes, M. D., Craigie, I. D., Harrison, L. B., Geldmann, J., Collen, B., Whitmee, S., ... Woodley, S. Wildlife population trends in protected areas predicted by national socio-economic metrics and body size. *Nat. Commun.* **7**, 12747 (2016).

Gray, C. L., Hill, S. L. L., Newbold, T., Hudson, L. N., Börger, L., Contu, S., ... Scharlemann, J. P. W. Local biodiversity is higher inside than outside terrestrial protected areas worldwide. *Nat. Commun.* **7**, 12306 (2016).

Reviewers' comments second round:

Reviewer #1 (Remarks to the Author):

The authors have substantially revised the manuscript and addressed many of the concerns (although not all – see below). The interpretation of the results is now more modest and in line with findings (for the most part – see below), and some new analyses reveal some surprising differences. However, the new narrative and organization of the paper undermines the findings and leaves me with uncertainty about major take-aways from the study. For example, the abstract seems to doubt the study findings. Perhaps the authors over-corrected in their revision.

In this revision, I don't find the framing around a methodological advancement particularly innovative, even if it is the first time matching and hurdle mixed-effects models are being used together for analyzing PAs. The narrative in the intro or discussion doesn't convince me with the depth or sophistication needed why matching is better or more advanced than other methods.

Most importantly (to me, at least), the authors haven't clearly presented and explained their results in a satisfying narrative. The story of the results unfortunately isn't yet coming through in a clear or exciting way.

RESULTS/DISCUSSION

It would be helpful to immediately follow a results statement with an interpretation of what it means for the study question. These are currently separated, which is disorienting and doesn't make good use of this unique section format offered by Nature Communications. For example, the authors report opposite results for bears (L223), but do not explain these results until pages later (L298). Interpretation immediately following such results would answer the questions that will be on readers minds. In other words, this type of combined Results/Discussion format enables the writers to explain the story as they share the results, as if explaining the results verbally to a group of smart non-specialists.

The whole Results/Discussion section would be stronger if the authors, at the very start of this section, framed the major findings and significance with a broad, brief overview. Then, in the next part of the section, dive into presenting results and interpreting them immediately thereafter. This could be followed by a broader discussion of the implications. Then follow with caveats and limitations. Be sure to conclude again by reiterating the major take-aways and significance. The current framing and organization doesn't yet convey a clean, crisp story.

I appreciate that the authors have woven into the Discussion some of the concerns raised by the three reviewers. However, the reviewers feedback should not drive the narrative of the story. The last paragraph of the Results/Discussion starting on L318 concludes the major body of the paper with a series of questions, representing doubts and caveats, which I find unsettling and which undermines the study findings.

Also, in L318, the authors state again, despite repeated requests from the reviewers not to over-generalize, that "our results revealed no clear differences in large carnivore densities between protected and non-protected sites in Finland", despite the mixed-effects model (admittedly one of two analyses) suggesting that bears do better in PAs. At the very least, qualifying that sentence (and any others I may have missed) to be factually accurate, such as "our results generally reveal a lack of differences..."

RESULTS

Results are not precisely stated: L166 states that "an upward trend...seems visible for the lynx", but this is not statistically significant (statistical tests are performed to avoid making speculations on trends based on visible patterns).

The narrative around interpretation of the results need to be made more streamline, concise and conclusive, especially L269-273.

GENERAL

Although the authors wrote in their rebuttal that they addressed some of my comments in the revision, the revised manuscript does not reflect those changes. Three examples:

L97: Latin names still do not have parentheses around them.

Use of "triangle" instead of "unit" in supplemental materials and Methods, and in some places in the main Results/Discussion text (L306).

Statistical test results are not provided for all results in the main text (e.g., L165 onward, L175 onward). As I wrote before, please report the values of the statistical tests when you refer to these results in the main text. If there are too many tests to report, include the range of results to showcase the evidence for your statements.

Reviewer #2 (Remarks to the Author):

The authors have done a fine job of addressing the substantial number of reviewer comments and overall, I appreciate the effort to provide a more nuanced interpretation of the results. I'll admit that I'm still not sure the matching analysis approach is an ideal way to assess PA effectiveness for large carnivores that are no doubt regularly moving into and out of PAs (judging from Table S1, only the Lapland PAs are likely large enough to contain even a single wolf home range). Perhaps this is why this method has only been used for forest cover in the past. It seems that a network of PAs and the linkages between them are what's important for any large carnivore population, and thus that the growth rates of populations/management units would be a better measure of overall PA network effectiveness (rather than the proportion of time spent inside of PAs, which is what I suspect the track counts are actually measuring). However, the paper seems methodologically sound, and I'll leave these broader considerations of whether the approach actually tells us much about PA effectiveness up to the discretion of the authors and the eventual reader. I have a few minor considerations below.

The abstract repeatedly refers to methods that have not be previously described. For instance, lines 39-40: "Some patterns for species, regions or time are detected only when combining the two analytical approaches and looking at interactions." Which two analytical approaches?

Lines 114-115: If a single individual carnivore is roaming into and out of a (relatively small) PA, perhaps it doesn't make sense to expect differences in population density inside and outside. Would the metric assessed by wildlife triangles be more appropriately stated as carnivore detections or activity?

To what degree does temporal activity play a role in these estimates of absolute PA effect, particularly for small PAs or for wildlife triangles located close to PA borders. A substantial amount of previous large carnivore work has demonstrated use of human-dominated landscapes predominantly during the night when human activity is low. One could imagine a similar situation here with larger carnivores using non-PA areas at night and then returning to PAs during the day as a refuge from people. Such a situation would obviously lead to similar levels of track accumulation inside and outside of PAs.

Lines 199-211: It's not clear to me what this paragraph is trying to convey. Have the authors concluded here that PAs have failed in some specific way (i.e., too much illegal hunting) that needs to be address by greater enforcement? If so, it's not clear how this conclusion was reached.

Lines 364-366: I thought the authors said that PA size was included in analyses. And given that Fig 1 shows polygons representing all PAs, who is it that size was not available?

Lines 452-457: It's still not clear to me why PA size wasn't included in the full model selection procedure described above. Couldn't areas of all PAs be calculated using the GIS polygons?

Fig. 2. There are not axis labels on this figure.

Point by point response to reviewers

Reviewers' comments:

Reviewer #1 (Remarks to the Author):

The authors have substantially revised the manuscript and addressed many of the concerns (although not all – see below). The interpretation of the results is now more modest and in line with findings (for the most part – see below), and some new analyses reveal some surprising differences. However, the new narrative and organization of the paper undermines the findings and leaves me with uncertainty about major take-aways from the study. For example, the abstract seems to doubt the study findings. Perhaps the authors over-corrected in their revision.

In this revision, I don't find the framing around a methodological advancement particularly innovative, even if it is the first time matching and hurdle mixed-effects models are being used together for analyzing PAs. The narrative in the intro or discussion doesn't convince me with the depth or sophistication needed why matching is better or more advanced than other methods.

Most importantly (to me, at least), the authors haven't clearly presented and explained their results in a satisfying narrative. The story of the results unfortunately isn't yet coming through in a clear or exciting way.

Response: We thank the reviewer for this comment. Our text was the result of finding a compromise between the comments of three reviewers, with the overall message that the main text should be toned-down. This explains why the reviewer does not find the new narrative as bold or innovative than the previous one. Yet we do believe that the results are novel at many fronts and a level of boldness is needed. We have made our best to improve this point throughout the text. For example we have restructured the whole Results/Discussion section following the advices provided below which should strengthen the narrative of this new version.

We have also modified the abstract to highlight the novelty of our approach and the relevance of the results presented in this study. See lines 33-42 p2.

RESULTS/DISCUSSION

It would be helpful to immediately follow a results statement with an interpretation of what it means for the study question. These are currently separated, which is disorienting and doesn't make good use of this unique section format offered by Nature Communications. For example, the authors report opposite results for bears (L223), but do not explain these results until pages later (L298). Interpretation immediately following such results would answer the questions that will be on readers minds. In other words, this type of combined Results/Discussion format enables the writers to explain the story as they share the results, as if explaining the results verbally to a group of smart non-specialists.

The whole Results/Discussion section would be stronger if the authors, at the very start of this section, framed the major findings and significance with a broad, brief overview. Then, in the next part of the section, dive into presenting results and interpreting them immediately thereafter. This could be followed by a broader discussion of the implications. Then follow with caveats and limitations. Be sure to conclude again by reiterating the major take-aways and significance. The current framing and organization doesn't yet convey a clean, crisp story.

Response: We thank the reviewer for this comment and the advices regarding a more suitable structure of the main text. We have now restructured the whole Results/Discussion section following the structure highlighted above. See lines 134 p 5 to line 321 p 14.

I appreciate that the authors have woven into the Discussion some of the concerns raised by the three reviewers. However, the reviewers feedback should not drive the narrative of the story. The last paragraph of the Results/Discussion starting on L318 concludes the major body of the paper with a series of questions, representing doubts and caveats, which I find unsettling and which undermines the study findings.

Response: We thank the reviewer for this comment. We have now removed this part including the series of questions and made the Conclusion more straightforward while highlighting at the same time the novelty of our approach and the importance of our results for improving PA targets (see lines 303-321 p 14).

Also, in L318, the authors state again, despite repeated requests from the reviewers not to over-generalize, that "our results revealed no clear differences in large carnivore densities between protected and non-protected sites in Finland", despite the mixed-effects model (admittedly one of two analyses) suggesting that bears do better in PAs. At the very least, qualifying that sentence (and any others I may have missed) to be factually accurate, such as "our results generally reveal a lack of differences..."

Response: We thank the reviewer for pointing out this discrepancy. We have used qualifiers to denote non-absolute generalities in our findings. The sentence is now rephrased (see line 303 p 14) and we revised the text throughout to avoid including such overstatements.

RESULTS

Results are not precisely stated: L166 states that "an upward trend...seems visible for the lynx", but this is not statistically significant (statistical tests are performed to avoid making speculations on trends based on visible patterns).

Response: We thank the reviewer for pointing out this imprecision and we agree that there are no significant differences in carnivore densities between protected and non-protected sites in any of the individual years. We have now deleted this part of the sentence.

The narrative around interpretation of the results need to be made more streamline, concise and conclusive, especially L269-273.

Response: We thank the reviewer for this suggestion. We have modified this paragraph to make the interpretation of the results more straightforward clear and concise. See lines 252-274 p 11-12.

GENERAL

Although the authors wrote in their rebuttal that they addressed some of my comments in the revision, the revised manuscript does not reflect those changes. Three examples:

Response: We apologize for having missed these caveats, the corrections of which should have been rather straightforward. We have now revised the manuscript accordingly.

L97: Latin names still do not have parentheses around them.

Response: All latin names have now parentheses around them. See lines 97-98 p 4 and line 235 p 10.

Use of “triangle” instead of “unit” in supplemental materials and Methods, and in some places in the main Results/Discussion text (L306).

Response: We have made sure to replace all “triangles” by “units” in the latest version of the MS and the Supplementary Material.

Statistical test results are not provided for all results in the main text (e.g., L165 onward, L175 onward). As I wrote before, please report the values of the statistical tests when you refer to these results in the main text. If there are too many tests to report, include the range of results to showcase the evidence for your statements.

Response: We thank the reviewer for this suggestion. Regarding the first part of the Results describing temporal variations in PA effectiveness: we have now added a short paragraph describing the range of median absolute PA effect over the study period for each carnivore species in addition to the Bonferroni-adjusted p-values that are always equal to 1. See lines 167-172 p7. Regarding the second part describing spatial variations in PA effectiveness, it would not make biological sense to report the range of median absolute PA effect for each region, by pooling the values of different carnivore species together (see Table S5. Supporting Material), therefore we added information about the fact that all 95% CI overlap 0. See line 184 p8.

Reviewer #2 (Remarks to the Author):

The authors have done a fine job of addressing the substantial number of reviewer comments and overall, I appreciate the effort to provide a more nuanced interpretation of the results. I'll admit that I'm still not sure the matching analysis approach is an ideal way to assess PA effectiveness for large carnivores that are no doubt regularly moving into and out

of PAs (judging from Table S1, only the Lapland PAs are likely large enough to contain even a single wolf home range). Perhaps this is why this method has only been used for forest cover in the past. It seems that a network of PAs and the linkages between them are what's important for any large carnivore population, and thus that the growth rates of populations/management units would be a better measure of overall PA network effectiveness (rather than the proportion of time spent inside of PAs, which is what I suspect the track counts are actually measuring). However, the paper seems methodologically sound, and I'll leave these broader considerations of whether the approach actually tells us much about PA effectiveness up to the discretion of the authors and the eventual reader. I have a few minor considerations below.

Response: We thank the reviewer for this comment. We do understand the concerns raised by the reviewer related to large carnivore ecology and the difficulties to assess PA effectiveness in maintaining these species. We note that the methodology developed for the Wildlife Triangle Scheme has been widely used and approved for large-scale biodiversity monitoring and to estimate mammal densities in northern Europe (Helle et al. 2016; Turkia et al. 2018; Wikenros et al. 2017). Moreover, snow track counts have also been widely used to monitor the density of several large carnivore species in the northern hemisphere: the Amur tiger (Hayward et al. 2002); the grey wolf (Kojola et al. 2014) and the wolverine (Golden et al. 2007). Therefore, we are confident about the robustness of the density indices used here as such (and not as proxies of the proportion of time spent inside PAs by large carnivores) and derived from the snow track counts. We agree that whether matching approaches are worth considering when evaluating PA effectiveness for largely mobile species remains a matter of debate and proper interpretation. Yet applied with proper data and in combination with other models, as done here, we believe matching approaches are a valid approach for population trends and not only for deforestation rates. Their limitation will largely be that of data availability. Thus we are convinced that the novel methodological approach used in this study represents an important methodological advance in terms of assessing PA effectiveness, being the first application of matching methods to wildlife data.

Golden, H. N., J. D. Henry, E. F. Becker, M. I. Goldstein, J. M. Morton, D. Sr. Frost, and A. J. Poe. 2007. Estimating wolverine *Gulo gulo* population size using quadrat sampling of tracks in snow. *Wildlife Biology* 13(Suppl. 2): 52–61.

Hayward GD, Miquelle DG, Smirnov EN, Nations C (2002) Monitoring Amur tiger populations: characteristics of track surveys in snow. *Wildl Soc Bull* 30(4): 1150–1159.

Helle, P., K. Ikonen, A. Kantola. Wildlife monitoring in Finland: online information for game administration, hunters, and the wider public. *Can. J. For. Res.*, 46 (12) (2016), pp. 1491-1496.

Kojola, I., Helle, P., Heikkinen, S., Lindén, H., Paasivaara, A., & Wikman, M. (2014). Tracks in snow and population size estimation: The wolf *Canis lupus* in Finland. *Wildlife Biology*, 20, 279–284.

Turkia T, Selonen V, Danilov P, Kurhinen J, Ovaskainen O, Rintala J, Brommer JE (2018b) Red squirrels decline in abundance in the boreal forests of Finland and NW Russia. *Ecography* 41:1370–1379.

Wikenros C, Aronsson M, Liberg O, Jarnemo A, Hansson J, Wallgren M, Sand H, Bergström R. 2017. Fear or food—abundance of red fox in relation to occurrence of lynx and wolf. *Sci Rep* 7:9059.

The abstract repeatedly refers to methods that have not be previously described. For instance, lines 39-40: “Some patterns for species, regions or time are detected only when combining the two analytical approaches and looking at interactions.” Which two analytical approaches?

Response: We thank the reviewer for pointing out this omission. We added one sentence on line 37 p2 specifying that we use a combination of matching methods and hurdle-mixed models to assess PA effectiveness. This is now described before the sentence highlighted by the reviewer in the abstract.

Lines 114-115: If a single individual carnivore is roaming into and out of a (relatively small) PA, perhaps it doesn't make sense to expect differences in population density inside and outside. Would the metric assessed by wildlife triangles be more appropriately stated as carnivore detections or activity?

Response: We thank the reviewer for this comment. We would not rename the metric assessed by wildlife triangles as carnivore detection or activity as we do not think that the scale issue pointed out by the reviewer here could be problematic regarding the interpretation of our results.

As stated above, snow track counts have been widely used to monitor the density of several large carnivore species in the northern hemisphere: the Amur tiger (Hayward et al. 2002); the grey wolf (Kojola et al. 2014) and the wolverine (Golden et al. 2007). Therefore, we are confident about the robustness of the density indices used here as such (and not as proxies of the proportion of time spent inside PAs by large carnivores) and derived from the snow track counts

In addition, both analytical methods use the total amount of data collected throughout the country over the study period in order to assess the effect of protection status on carnivore densities at the network level (comparing all protected wildlife units versus non-protected units) and not at the individual PA level.

To what degree does temporal activity play a role in these estimates of absolute PA effect, particularly for small PAs or for wildlife triangles located close to PA borders. A substantial

amount of previous large carnivore work has demonstrated use of human-dominated landscapes predominantly during the night when human activity is low. One could imagine a similar situation here with larger carnivores using non-PA areas at night and then returning to PAs during the day as a refuge from people. Such a situation would obviously lead to similar levels of track accumulation inside and outside of PAs.

Response: We thank the reviewer for this suggestion, however we do not see how variations in temporal activity would influence our estimates of absolute PA effect at the network level. We agree that large carnivores could indeed shift from protected to non-protected areas between day and night depending on levels of human disturbance and spatial variations in prey abundance as it has been demonstrated in other systems (see Filla et al. 2017). For example, one would expect that large carnivores use more protected areas during the day when human activity is high and select non-protected areas where ungulates are often more abundant during the night when human activity is low. Our data cannot detect such fine-scale movements inside and outside PAs but this was not the objective of this study and such patterns should be studied using GPS telemetry.

The triangle survey reflects carnivore densities over several days. It could of course vary with snow freshness, temperature and wind but in theory footprints on snow can show data over day and night and thus if carnivores roam in and out of PAs, their presence will be detected in both areas. Therefore data collected in protected and unprotected sites are comparable and there is no bias linked to variations in temporal activity at the network scale.

Lines 199-211: It's not clear to me what this paragraph is trying to convey. Have the authors concluded here that PAs have failed in some specific way (i.e., too much illegal hunting) that needs to be address by greater enforcement? If so, it's not clear how this conclusion was reached.

Response: We agree that this paragraph was too speculative regarding recommendations for future research. We have considerably shortened this part of the text. See lines 271-274 p 13.

Lines 364-366: I thought the authors said that PA size was included in analyses. And given that Fig 1 shows polygons representing all PAs, who is it that size was not available?

Response: We thank the reviewer for pointing out this omission. We forgot to modify this part of the text after the last revision (after adding PA size as a covariate in the hurdle-mixed models). The sentence has now been modified (see lines 352-354 p 16).

Lines 452-457: It's still not clear to me why PA size wasn't included in the full model selection procedure described above. Couldn't areas of all PAs be calculated using the GIS polygons?

Response: We thank the reviewer for this comment. We have first done as the reviewer suggested above, i.e. including PA size in the full model selection procedure. However the hurdle-mixed models did not converge when the two covariates ‘protected area size’ and ‘site protection status’ (protected/non protected) were included in the same models (in addition to the 8 other covariates). The only option we found to get around this problem was to create an additional set of models specifically testing the effect of PA size on the density of the 4 carnivore species studied here, in addition to the other 8 covariates.

Fig. 2. There are not axis labels on this figure.

Response: We thank the reviewer for pointing out this omission. We have now added axis labels on Figure 2. See page 8.